# Videos are Sample-Efficient Supervisions: Behavior Cloning from Videos via Latent Representations

**Xin Liu**[1,2], **Haoran Li**[1,2,*], **Dongbin Zhao**[1,2]
[1]State Key Laboratory of Multimodal Artificial Intelligence Systems,
Institute of Automation, Chinese Academy of Sciences
[2]School of Artificial Intelligence, University of Chinese Academy of Sciences
{liuxin2021,lihaoran2015,dongbin.zhao}@ia.ac.cn

## Abstract

Humans can efficiently extract knowledge and learn skills from the videos within only a few trials and errors. However, it poses a big challenge to replicate this learning process for autonomous agents, due to the complexity of visual input, the absence of action or reward signals, and the limitations of interaction steps. In this paper, we propose a novel, unsupervised, and sample-efficient framework to achieve imitation learning from videos (ILV), named Behavior Cloning from Videos via Latent Representations (BCV-LR). BCV-LR extracts action-related latent features from high-dimensional video inputs through self-supervised tasks, and then leverages a dynamics-based unsupervised objective to predict latent actions between consecutive frames. The pre-trained latent actions are fine-tuned and efficiently aligned to the real action space online (with collected interactions) for policy behavior cloning. The cloned policy in turn enriches the agent experience for further latent action finetuning, resulting in an iterative policy improvement that is highly sample-efficient. We conduct extensive experiments on a set of challenging visual tasks, including both discrete control and continuous control. BCV-LR enables effective (even expert-level on some tasks) policy performance with only a few interactions, surpassing state-of-the-art ILV baselines and reinforcement learning methods (provided with environmental rewards) in terms of sample efficiency across 24/28 tasks. To the best of our knowledge, this work for the first time demonstrates that videos can support extremely sample-efficient visual policy learning, without the need to access any other expert supervision.

## 1  Introduction

Reinforcement learning (RL) [1, 2, 3] has demonstrated powerful capabilities in solving decision-making tasks across different fields [4, 5, 6, 7]. However, the stringent training conditions, including well-designed rewards and substantial environmental interactions, have greatly restricted the application scenarios of RL [8, 9, 10]. Behavior cloning from action-labeled expert trajectories [11, 12] or offline RL from exploratory training experience [13, 14] provide solutions for policy learning without any environmental interactions, which means the highest sample efficiency. Nevertheless, the required offline supervision of high quality is generally not easily accessible.

Compared with well-designed rewards or qualified offline datasets, videos are a kind of supervisory information that is much easier to obtain. Imitating skills from expert videos is also a natural and efficient learning method for humans, e.g., playing games by watching tutorial videos on the Internet. However, in contrast to humans' ease in learning from videos, it is no simple feat to replicate this

---

*Corresponding author.

39th Conference on Neural Information Processing Systems (NeurIPS 2025).

efficient watch-and-learn process for autonomous agents, which are mainly attributed to the complex visual input and missed action labels. Although numerous studies have already utilized videos as a valuable supplement to boost policy learning from expert rewards [15, 16, 17, 18] or expert actions [19, 20, 21], imitating entirely from videos without any other supervision is a more natural and ideal learning mode.

Formally, this problem is named Imitation Learning from Videos (ILV). Currently, the mainstream approach to solving ILV is inverse RL [22, 23, 24]. These methods aim to extract reward signals from videos that are highly consistent with expert policies, and then perform imitation through RL. However, the training of an extra reward prediction network and the RL value network both necessitate extensive exploration of the observation space, which inevitably leads to lower sample efficiency than traditional RL methods (based on steady expert rewards) in many cases [25]. Another category of approaches [26, 27] aims to predict the missed expert actions based on environmental interactions, conducting supervised learning (i.e., behavioral cloning) to imitate policies [28, 29]. Such methods require less knowledge of environment dynamics [25], demonstrating the potential to reduce reliance on environmental samples [26, 27] in state-based imitation. However, when it comes to video imitation, both the difficulty of interpreting observations and predicting expert actions increase sharply. Existing supervised imitation methods often encounter performance bottlenecks and are unable to learn policies close to the expert level [28].

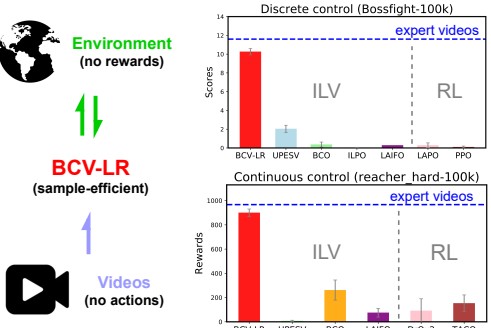

Figure 1: BCV-LR achieves sample-efficient video-based imitation learning without accessing expert actions or rewards. It achieves expert-level policy performance on discrete task "Bossfight" and continuous task "reacher_hard" with only 100k interactions allowed, surpassing state-of-the-art ILV and RL baselines.

The above discussion naturally leads to a question: Is it possible to balance the effectiveness and sample efficiency in visual policy learning, where only videos are accessible supervisions? In this paper, we try to answer this question through Behavior Cloning from Videos via Latent Representations (BCV-LR), which conducts offline latent pre-training to extract rich knowledge from videos for efficient adaptation to real environments online. Concretely, BCV-LR contains an offline pre-training stage and an online finetuning stage. In offline pre-training, BCV-LR first pre-trains a self-supervised visual encoder over the video, aiming to extract the action-related information from raw pixels and thus alleviating the learning difficulty in subsequent training. Based on the latent features, BCV-LR employs another trainable world model, optimizing a dynamics-based objective in an unsupervised manner, to obtain the latent actions between consecutive video frames. In the online stage, BCV-LR fine-tunes the latent actions with the pretrained world model over the collected reward-free transitions, simultaneously aligning these implicit actions to the real action space for behavior cloning of a policy. The cloned policy in turn enriches the collected experience, leading to better latent action finetuning and decoding, which finally results in an iterative policy improvement that is extremely sample-efficient.

We conduct extensive experiments on a set of challenging visual control tasks, including 16 discrete control tasks from the Procgen benchmark [30] and 12 continuous control tasks from the Deepmind Control suite (DMControl) [31] and Metaworld [32]. Even with only a few environmental interactions permitted, BCV-LR can still enable effective and even expert-level policy learning performance on many tasks, which both recent advanced ILV baselines (provided with expert videos) and RL methods (provided with expert rewards) cannot achieve, as shown in Figure 1. These indicate the state-of-the-art sample efficiency of the proposed BCV-LR. To the best of our knowledge, the proposed BCV-LR for the first time demonstrates the feasibility of using videos as the only expert supervisory signal to guide extremely sample-efficient visual policy learning. We provide the implementation of BCV-LR at `https://github.com/liuxin0824/BCV-LR`.

We summarize the contributions of this paper as follows:

- We propose a novel framework for sample-efficient ILV, named Behavior Cloning from Videos via Latent Representations (BCV-LR). To the best of our knowledge, our work for the first time demonstrates that videos can support extremely sample-efficient visual policy learning, without the need for expert actions or rewards.

- BCV-LR extracts action-related latent features from high-dimensional video frames through self-supervised tasks, and then leverages a dynamics-based unsupervised objective to predict latent actions between consecutive images. With online interactions, the pre-trained latent actions are fine-tuned and aligned to the real action space for behavior cloning. The cloned policy in turn enriches the agent experience for better latent action finetuning, resulting in an iterative policy improvement that is highly sample-efficient.

- We conduct extensive experiments on a set of challenging visual control tasks, including both discrete control and continuous control. The results demonstrate that the proposed BCV-LR exhibits state-of-the-art sample efficiency, surpassing recent advanced online visual policy learning baselines, including both ILV methods and RL methods.

## 2 Related Works

### 2.1 Sample-Efficient Visual Policy Learning

Interacting with the environment to collect data is an expensive and dangerous process in many scenarios [33, 8], which makes sample efficiency an important metric for policy learning methods [34, 28]. Self-supervised RL [35, 36, 37, 38] employs extra auxiliary tasks in addition to the RL objective, accelerating downstream policy learning by improving visual understanding capabilities in a target manner. Model-based RL methods [39, 40, 41, 42, 43] train extra world models that augment the RL experience, thereby reducing the requirement of real interactions [44, 45, 46]. These methods indeed achieve higher sample efficiency, but they are inherently limited by the non-trivial hand-craft rewards [47]. At the same time, some approaches try to completely avoid accessing environments, such as conducting offline RL over the training experience [14, 13, 15] and behavior cloning with action-labeled expert demonstrations [12, 11]. However, these offline expert datasets of high quality can be expensive and even unavailable in many scenarios.

### 2.2 Utilizing Videos as Supervisions

With the continuous development of the Internet, obtaining a vast number of diverse videos has become extremely convenient nowadays. Many of these videos contain expert-level demonstrations, which serve as a valuable source of knowledge for both humans and machines. However, due to the lack of action information, it poses a significant challenge to well utilize these videos in machine learning. One popular way is to improve existing strategy learning methods with the help of videos, such as video-based pre-training for RL [48, 49, 21], video-based intrinsic reward for RL [17, 50], and video-based behavior cloning with expert actions [19, 20]. In addition, FICC [51] pre-trains world models on action-free agent experience, accelerating model-based RL on Atari games. JPET [52] utilizes a mixed dataset containing videos to achieve one-shot visual imitation. Although these methods have achieved certain success, they still require dependence on other expert supervisions. In contrast, ILV [22, 47, 50, 28], which aims to acquire skills from only videos in a manner similar to humans, has a much broader range of application scenarios.

### 2.3 Imitation Learning from Videos (ILV)

Imitation Learning from Observations (ILO) [23, 26, 27] is proposed to recover the expert policies hidden in the action-free demonstrations. As an extension of ILO to the visual domain, ILV poses a greater challenge due to the more complex visual inputs. The mainstream approaches to solving the ILV problem are inverse RL [50, 47, 53, 54, 55], which tries to extract and optimize the reward signals contained in the videos. They achieve considerable results on imitation performance but usually can't well balance sample efficiency [25, 28]. This conforms to our intuition, because both reward prediction and RL require advanced understanding of the environments, which cannot be achieved without sufficient interactions. At the same time, some researchers give up reward engineering, trying to predict expert actions from videos [26, 56, 27] for supervised ILO. They achieve considerable success but suffer from severe performance drops when transferred into more

complex visual continuous control [29, 28]. In this situation, BCV-LR for the first time (i) taps into the potential of supervised ILV and (ii) exhibits the feasibility of using videos as the only supervisory signal to guide highly sample-efficient visual policy learning.

## 3 Behavior Cloning from Videos via Latent Representations

**Problem Definition of ILV**   The goal of ILV is to imitate the policy contained in the expert videos by interacting with a reward-free environment. Concretely, the expert videos can be regarded as a dataset containing action-free expert transitions $\{(o_i^v, o_{i+1}^v)\}$. The observation $o$ can represent a single frame or stacked historical frames, depending on the partial observability of the specific task. The interactive environment can be denoted as a reward-free Markov Decision Process (MDP) $\mathcal{M} = (\mathcal{O}, \mathcal{A}, \mathcal{P}, d_0)$, where $\mathcal{O}$ is the visual observation space, $\mathcal{A}$ is the action space, $\mathcal{P}$ is the observation transition function, and $d_0$ is the distribution of the initial observation. Based on the reward-free environmental transitions $\{(o_t^e, a_t^e, o_{t+1}^e)\}$ and the videos $\{(o_i^v, o_{i+1}^v)\}$, the ILV agents can achieve policy learning by estimating the expert rewards or expert actions [28, 50].

**Framework of BCV-LR**   BCV-LR addresses ILV problem by estimating the expert actions contained in the videos. The predicted actions are used to obtain a policy through behavior cloning. BCV-LR contains two stages: an offline pre-training stage and an online finetuning stage.

In offline pre-training, BCV-LR first pre-trains a self-supervised visual encoder $f$ over the videos, aiming to extract the action-related information from raw pixels and thus alleviating the learning difficulty for both action prediction and policy cloning. Based on the pre-trained latent features, BCV-LR employs another trainable world model $w$ along with the latent action predictor $p$, optimizing a dynamics-based objective in an unsupervised manner. This aims to obtain the latent actions between consecutive video frames. In the online stage, BCV-LR fine-tunes the latent actions with the pretrained world model $w$ over the collected reward-free transitions, efficiently aligning these pseudo actions to the real action space via a latent action decoder $d$. The transitions are collected by imitating a latent policy $\pi$ that clones the latent actions based on latent features from expert videos. $\pi$ is combined with latent feature encoder $f$ and latent action decoder $d$ to interact with the environment. As latent actions are finetuned, the cloned latent policy improves simultaneously, which in turn collects transitions at a higher performance level for better latent action finetuning, resulting in an iterative policy improvement. After the online stage, $f$, $\pi$, and $d$ together form the final policy of BCV-LR. We provide a diagram in Figure 2 and pseudocode in Appendix A.

### 3.1 Offline Pretraining on Action-free Videos

To facilitate the utilization of environmental interactions in online learning, BCV-LR firstly extracts knowledge from videos offline. We describe how to pre-train a feature encoder $f$ to extract useful information from high-dimension videos in Section 3.1.1, and how to train a latent action predictor $p$ and a world model $w$ jointly through an unsupervised dynamics-based objective in Section 3.1.2.

#### 3.1.1 Learning Latent Features through Self-Supervised Tasks

To accurately predict the missed actions from the expert videos and support effective downstream policy, the feature encoder $f$ should be able to extract information related to decision-making from complex high-dimensional visual inputs, which is consistent with the requirements in visual RL. Inspired by the recent success of self-supervised RL [35, 57, 34], BCV-LR learns its feature encoder $f$ by optimizing self-supervised objectives defined on the expert videos. Note that the proposed BCV-LR is compatible with any action-free self-supervised tasks. By choosing appropriate self-supervised objectives, it can adapt to different types of visual control tasks easily. In addition, BCV-LR can also be combined with a well-trained, off-the-shelf encoder, which enables it to retain the potential for handling tasks with much more complex visual inputs.

For tasks with relatively simple dynamics but involving complex visual information (e.g., Procgen video games [30]), BCV-LR optimizes a contrastive learning objective to align two different randomly shifted images of the same observation together. This makes $f$ focus more on relative position difference that is highly related to actions. In practice, we empirically find that letting $f$ simultaneously achieve a self-reconstruction task yields further improvement in Procgen. Concretely, a batch of observations $\{o_i^v\}_{i=1}^N$ are sampled from videos. Each $o_i^v$ is randomly shifted twice to obtain two

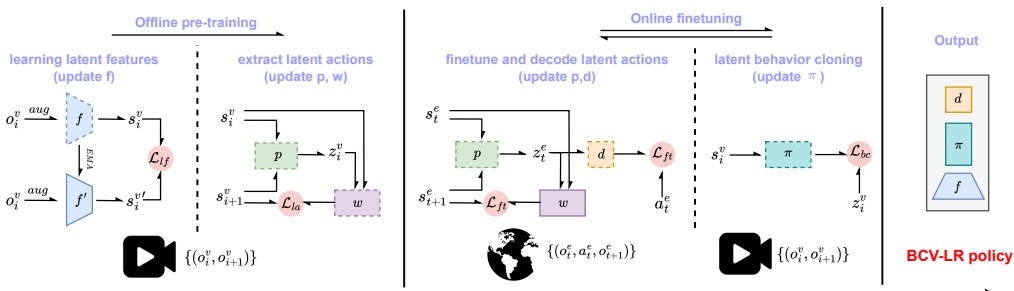

Figure 2: The training objectives of different stages. BCV-LR first pre-trains a self-supervised feature encoder $f$ over the videos. Based on the latent features, BCV-LR employs another trainable world model $w$ along with the latent action predictor $p$, optimizing a dynamics-based objective in an unsupervised manner to obtain the latent actions between consecutive video frames. In the online stage, BCV-LR fine-tunes the latent actions with the pretrained world model $w$ over the collected reward-free transitions, aligning latent actions to the real action space via a latent action decoder $d$. Simultaneously, BCV-LR trains a latent policy $\pi$ that clones the latent actions, which shares the latent feature encoder $f$ and latent action decoder $d$ to interact with the environment. This enriches collected data for further latent action finetuning, resulting in an iterative improvement. Note that $f$, $\pi$, and $d$ together form the final policy of BCV-LR.

augmented images $aug(o_i^v)$ and $aug'(o_i^v)$. They are then separately encoded by $f$ and its momentum encoder $f'$ (updated via Exponential Moving Average (EMA) [58]) to obtain two latent features, $s_i^v$ and $s_i^{v'}$. The self-supervised target contains a contrastive learning loss and a self-reconstruction loss:

$$\mathcal{L}_{lf} = -\log \frac{\exp(u(s_i^v)^\top W s_i^{v'})}{\sum_{j=1}^N \exp(u(s_i^v)^\top W s_j^{v'})} + \alpha ||v(s_i^v) - aug(o_i^v)||^2, \tag{1}$$

where $W$ is a trainable matrix employed in contrastive learning, $\alpha$ is a coefficient scaling two losses, $u(\cdot)$ is a Multi-Layer Perceptron (MLP) set to introduce asymmetry for avoiding collapse to trivial solutions, and $v(\cdot)$ is a trainable reconstruction decoder.

As previously mentioned, BCV-LR can select self-supervised tasks tailored to different domains to achieve better latent feature extraction. For partially observable domains with complex dynamics (e.g., continuous DMControl [31]), understanding the temporal information is proven crucial for the representation module to support effective downstream policy learning [59, 34, 60]. This motivates us to employ a recent advanced prototype-based temporal association task [57, 43] based on the Sinkhorn-Knopp [61] algorithm, which aligns two temporally neighboring observations together in the latent space. For the sake of fluency of the main text, we put the details of this self-supervised temporal association task in Appendix A.

### 3.1.2 Extracting Latent Actions over Latent Features

With the pre-trained encoder $f$, each pixel observation $o_i^v$ is encoded into the latent feature $s_i^v$. BCV-LR then trains a predictor $p$ to extract the latent action $z_i^v$ between neighboring feature pairs $(s_i^v, s_{i+1}^v)$. To obtain $z_i^v$ in an unsupervised manner, another world model $w$ is employed to reconstruct the next feature $s_{i+1}^v$ based on the current feature $s_i^v$ and the predicted $z_i^v$. The motivation is intuitive: the latent action should contain information that describes the changes between frames. To avoid the trivial solution that $p$ learns to naively copy the next latent features into latent actions, the continuous $z_i^v$ is discretized via vector quantization (VQ) [62] before entering the world model $w$ [28, 19]. Concretely, $z_i^v$ is mapped to its nearest vector $z_i^{vq}$ defined in a limited codebook, which forces the model to find commonalities among numerous transition pairs. The $z_i^{vq}$ will replace $z_i^v$ to achieve forward propagation in world model $w$, and the reconstruction loss can be defined as the following:

$$\mathcal{L}_{la} = ||w(s_i^v, z_i^{vq}) - s_{i+1}^v||^2. \tag{2}$$

In backpropagation, the gradient of $z_i^{vq}$ is copied to $z_i^v$ directly. $p$ and $w$ are optimized jointly by $\mathcal{L}_{la}$. In addition, $p$ is also updated along with the codebook in the VQ training process [21].

## 3.2 Online Finetuning with Reward-free Interactions

In the online learning stage, BCV-LR is able to quickly adapt the pre-trained knowledge to the real environment, thus achieving sample-efficient policy learning. In Section 3.2.1, we describe how to finetune and align the latent actions to real action space with the latent action decoder $d$, based on agent experience. By sharing $f$ and $d$, BCV-LR clones the latent actions via a latent policy $\pi$ to interact with the environments, shown in Section 3.2.2. This policy in turn enriches the agent experience, resulting in an iterative improvement of both latent action predictor $p$ and latent policy $\pi$. After the online stage, $f$, $\pi$, and $d$ together form the final policy of BCV-LR.

### 3.2.1 Finetuning and Decoding Latent Actions with Pre-trained World Model

BCV-LR interacts with the environment to collect environmental transitions with real actions $\{(o_t^e, a_t^e, o_{t+1}^e)\}$. By letting latent action predictor $p$ and decoder $d$ predict real actions over these environmental transitions, BCV-LR further finetunes $p$, while aligning the predicted latent actions to the real action space with decoder $d$. Considering that there are distribution differences between the collected transitions and expert videos, BCV-LR simultaneously utilizes the pre-trained world model $w$ to provide expert dynamics constraints. This forces the prediction model $p$ to maintain its understanding of environmental dynamics during adaptation to non-expert action labels, enhancing the finetuning robustness and achieving better results in practice. Concretely, each environmental observation pair $(o_t^e, o_{t+1}^e)$ is encoded by the pre-trained $f$ into the latent features $(s_t^e, s_{t+1}^e)$. They are further passed into $p$ to generate the predicted latent action $z_t^e$. The learning objective contains (i) the action prediction loss and (ii) the future reconstruction loss over the collected transitions:

$$\mathcal{L}_{ft} = -a_t^{eT} \log(\text{softmax}[d(z_t^e)]) + \beta ||w(s_t^e, z_t^{vq}) - s_{t+1}^e||^2, \tag{3}$$

where the $\beta$ is a coefficient scaling two losses and $z_t^{vq}$ is the codebook vector nearest to $z_t^e$. BCV-LR finetunes $p$ and learns $d$ simultaneously through backpropagation. $p$ and the codebook are also updated through VQ training. Finetuning $w$ is optional, leading to better performance on some tasks in practice. The cross-entropy loss is replaced with Mean Square Error (MSE) in continuous control.

### 3.2.2 Learning Latent Policy via Behavior Cloning

With the predicted actions aligned into the real action space, BCV-LR can imitate a policy directly executed in the environment from expert videos. By sharing the pre-trained feature encoder $f$ and the latent action decoder $d$, BCV-LR only needs to train a latent policy $\pi$ that maps the latent features into the latent actions. Concretely, an expert video transition $(o_i^v, o_{i+1}^v)$ is successively fed into feature encoder $f$ to produce $(s_i^v, s_{i+1}^v)$, and the latent action predictor $p$ to produce the predicted latent action $z_i^v$ as the label. The behavior cloning objective is correspondingly defined as:

$$\mathcal{L}_{bc} = ||\pi(s_i^v) - z_i^v||^2. \tag{4}$$

The $\pi$-output latent action is then decoded to the real action space by $d$ for execution. After the online stage, $f$, $\pi$, and $d$ together form the final policy of BCV-LR.

## 4 Experiments

In this section, we provide a comprehensive evaluation of the proposed BCV-LR. We first briefly introduce the experimental settings (Section 4.1), then provide the results of discrete control and continuous control (Section 4.2 & 4.3), conduct ablation studies (Section 4.4), and provide analytical experiments on video data efficiency and multi-task adaptation (Section 4.5 & 4.6). You can refer to Appendix B for more results and analysis, and Appendix C for detailed experimental settings.

### 4.1 Experimental Settings

#### 4.1.1 Environments

We test the performance of BCV-LR in a set of challenging domains with complex visual inputs and environment dynamics, including 16 discrete control tasks from Procgen benchmark [30] and 8

Table 1: The interaction-limited policy learning in Procgen. *Italics* indicate using expert videos, and underlines denote using environmental rewards. **Bold text** indicates the highest score excluding video experts. BCV-LR exhibits the highest sample efficiency, utilizing only videos.

| Task | BCV-LR (ours) | UPESV [28] | BCO [26] | ILPO [27] | LAIFO [50] | LAPO [21] | PPO [63] | Expert Videos |
|---|---|---|---|---|---|---|---|---|
| Bigfish | **35.9 ± 2.0** | 30.5 ± 1.6 | 3.6 ± 3.7 | 0.8 ± 0.1 | 0.8 ±0.0 | 20.6 ± 0.7 | 0.9 ± 0.1 | 36.3 |
| Maze | **9.9 ± 0.1** | 9.7 ± 0.2 | 7.4 ± 2.4 | 4.2 ± 0.3 | 4.3 ± 0.4 | 9.6 ± 0.1 | 5.0 ± 0.7 | 10.0 |
| Heist | 9.3 ± 0.1 | **9.4 ± 0.3** | 7.6 ± 1.9 | 6.7 ± 0.5 | 5.4 ± 0.6 | **9.4 ± 0.3** | 3.7 ± 0.2 | 9.7 |
| Coinrun | **8.9 ± 0.0** | 7.4 ± 0.2 | 6.7 ± 0.9 | 3.7 ± 1.3 | 4.7 ± 0.1 | 6.2 ± 0.4 | 4.1 ± 0.5 | 9.9 |
| Plunder | 4.4 ± 0.2 | 3.5 ± 0.7 | 4.2 ± 0.3 | 2.2 ± 1.3 | 3.2 ± 0.9 | **4.8 ± 0.1** | 4.4 ± 0.4 | 11.5 |
| Dodgeball | **12.4 ± 0.8** | 9.1 ± 0.8 | 5.4 ± 1.1 | 0.6 ± 0.1 | 0.8 ± 0.2 | 5.9 ± 1.1 | 1.1 ± 0.2 | 13.5 |
| Jumper | **7.5 ± 0.3** | 6.6 ± 0.2 | 6.4 ± 0.3 | 3.1 ± 0.6 | 3.9 ± 0.4 | 7.3 ± 0.2 | 3.5 ± 0.7 | 8.5 |
| Climber | **9.4 ± 0.6** | 6.8 ± 0.6 | 3.3 ± 0.2 | 3.4 ± 0.5 | 2.7 ± 0.9 | 4.7 ± 0.3 | 2.2 ± 0.2 | 10.2 |
| Fruitbot | **27.5 ± 1.5** | 20.6 ± 1.6 | 3.5 ± 0.5 | -2.0 ± 0.7 | -2.5 ± 0.1 | 0.5 ± 0.3 | -1.9 ± 1.0 | 29.9 |
| Starpilot | **54.8 ± 1.4** | 15.0 ± 0.8 | 12.8 ± 13.9 | 0.5 ± 0.7 | 2.0 ± 0.7 | 20.3 ± 1.6 | 2.6 ± 0.9 | 67.0 |
| Ninja | **7.2 ± 0.3** | 6.3 ± 0.3 | 4.2 ± 1.1 | 2.2 ± 1.1 | 3.0 ± 0.1 | 5.2 ± 0.1 | 3.4 ± 0.3 | 9.5 |
| Miner | **11.6 ± 0.2** | 9.3 ± 1.2 | 5.8 ± 1.3 | 1.2 ± 0.4 | 1.2 ± 0.2 | 6.7 ± 0.6 | 1.2 ± 0.2 | 11.9 |
| Caveflyer | **4.6 ± 0.2** | 3.5 ± 0.6 | 2.8 ± 1.1 | 3.2 ± 0.3 | 2.4 ± 0.9 | 3.9 ± 0.1 | 3.0 ± 0.4 | 9.2 |
| Leaper | **4.0 ± 0.2** | 2.9 ± 0.3 | 2.5 ± 0.5 | 2.6 ± 0.2 | 1.9 ± 0.2 | 2.7 ± 0.2 | 2.6 ± 0.3 | 7.4 |
| Chaser | **3.1 ± 0.5** | 0.8 ± 0.1 | 0.8 ± 0.0 | 0.7 ± 0.0 | 0.6 ± 0.1 | 0.8 ± 0.0 | 0.4 ± 0.2 | 10.0 |
| Bossfight | **10.3 ± 0.3** | 2.0 ± 0.4 | 0.4 ± 0.3 | 0.1 ± 0.0 | 0.3 ± 0.0 | 0.3 ± 0.3 | 0.1 ± 0.1 | 11.6 |
| Mean | **13.8** | 9.0 | 4.8 | 2.1 | 2.2 | 6.8 | 2.3 | 16.6 |
| Video-norm Mean | **0.79** | 0.58 | 0.38 | 0.22 | 0.22 | 0.48 | 0.22 | 1.00 |

continuous control tasks from Deepmind Control suite (DMControl) [31]. Procgen [30] provides a diverse set of procedurally generated video game environments, each with unique challenges, different dynamics, and especially changing visual styles. DMControl [31] provides a series of challenging robot control tasks. Although the visual changes are not as complex as those in Procgen, the continuity of the observation and action spaces, as well as the partial observability of the states, greatly increase the difficulty of understanding the environmental dynamics. The complex visual inputs and environmental dynamics make these two benchmarks extremely challenging for ILV problems, which is recognized and widely employed by recent advanced studies [21, 50, 54, 28, 47]. In addition, we also provide results on 4 robotic manipulation tasks from Metaworld [32] to demonstrate a wider application of BCV-LR, which is detailed in Appendix B.

### 4.1.2 Baselines

We employ several popular and advanced ILV baselines: UPESV [28], LAIFO [50], ILPO [27], and BCO [26]. LAIFO is a recent advanced inverse RL approach that derives rewards from the expert videos through adversarial imitation techniques. The other three methods all try to recover the missed actions from the expert observations (videos) and obtain the imitated policies via behavior cloning, which is similar to the proposed BCV-LR. Among them, UPESV and LAIFO have achieved state-of-the-art ILV performance in discrete Procgen and continuous DMControl, respectively.

To provide a comprehensive comparison, we further employ several online RL baselines provided with environmental rewards. The advanced RL algorithms vary across different benchmarks. For discrete tasks, LAPO [21] and PPO [63] are employed. PPO is the most popular RL baseline in Procgen, while LAPO further leverages expert videos to improve PPO by pre-training latent policy. For continuous tasks, we employ DrQv2 [3] and TACO [60]. DrQv2, which combines several image-oriented tricks with DDPG, is currently the most popular RL method for continuous visual control. TACO is a state-of-the-art self-supervised RL method, which improves DrQv2 via several visual auxiliary objectives. All of these baselines are recognized as leading RL methods in the corresponding benchmarks.

Note that the ILV methods, including the proposed BCV-LR and four baselines (UPESV, BCO, ILPO, and LAIFO), can only access expert videos. We directly compare the RL methods and the ILV methods in terms of online sample efficiency, but in fact, they rely on different supervisory information.

### 4.2 Discrete Control

In this section, we test the proposed BCV-LR in 16 discrete control tasks from Procgen. We choose "easy" mode and "full" distributions of levels across all tasks. On each task, only 100k environmental

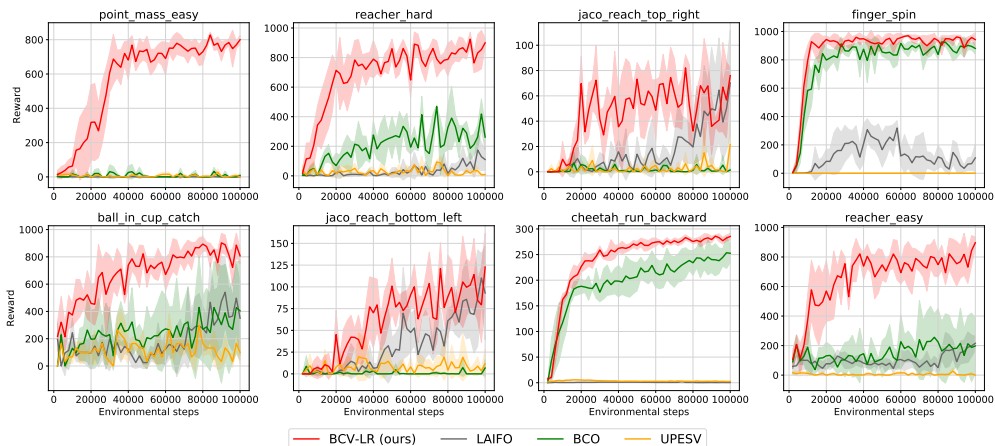

Figure 3: Online training curves of ILV methods in DMControl. BCV-LR can efficiently utilize environmental samples and learn effective strategies at 50k steps (even 20k on some tasks).

Table 2: Comparison with both ILV and RL baselines on DMControl. *Italics* indicate using expert videos, and underlines denote using environmental rewards. **Bold text** indicates the highest score excluding video experts. BCV-LR exhibits the leading sample efficiency under supervisions of only videos.

| Task | *BCV-LR (ours)* | *LAIFO* [50] | *BCO* [26] | *UPESV* [28] | TACO [60] | DrQv2 [3] | Expert Videos |
|---|---|---|---|---|---|---|---|
| point_mass_easy | **800 ± 25** | 0 ± 0 | 8 ± 12 | 0 ± 0 | 712 ± 96 | 525 ± 189 | 885 |
| reacher_hard | **900 ± 31** | 110 ± 71 | 262 ± 83 | 8 ± 5 | 154 ± 68 | 92 ± 98 | 967 |
| jaco_reach_top_right | 76 ± 6 | 70 ± 41 | 1 ± 2 | 22 ± 17 | **80 ± 12** | 37 ± 9 | 198 |
| finger_spin | **942 ± 48** | 108 ± 96 | 881 ± 39 | 0 ± 0 | 618 ± 69 | 374 ± 264 | 981 |
| ball_in_up_catch | **807 ± 57** | 350 ± 271 | 404 ± 363 | 100 ± 81 | 321 ± 166 | 246 ± 137 | 986 |
| jaco_reach_bottom_left | **123 ± 39** | 93 ± 54 | 7 ± 10 | 12 ± 16 | 46 ± 9 | 23 ± 10 | 203 |
| cheetah_run_backward | 285 ± 6 | 0 ± 0 | 253 ± 18 | 0 ± 0 | 304 ± 175 | **332 ± 12** | 389 |
| reacher_easy | **897 ± 38** | 215 ± 47 | 198 ± 189 | 0 ± 0 | 244 ± 98 | 225 ± 91 | 975 |
| Mean | **604** | 158 | 336 | 18 | 310 | 232 | 698 |
| Video-norm Mean | **0.78** | 0.20 | 0.31 | 0.03 | 0.45 | 0.34 | 1.00 |

steps are allowed. This number is much smaller than 4M or 25M employed by advanced RL works [30, 64, 21], requiring extremely high sample efficiency. The expert video dataset containing 8M steps is generated by well-trained RL agents, provided by [21]. For RL methods, we allow them to access the expert reward provided by Procgen environments. These all follow recent advanced works [21, 28]. Refer to Appendix C for detailed hyper-parameter settings.

The results are shown in Table 1. Compared with ILV methods based on behavior cloning (UPESV, BCO, and ILPO), the proposed BCV-LR achieves consistent leadership across all 16 tasks, indicating a more accurate and robust action prediction for better behavior cloning. By contrast, the inverse RL method LAIFO shows less satisfactory performance. Given that the diverse visual styles of the Procgen pose significant challenges for reward prediction [54, 53], we directly make comparisons with state-of-the-art RL methods accessing ground-truth environmental rewards, and BCV-LR still demonstrates a clear advantage. The results fully demonstrate the superiority of BCV-LR in terms of sample efficiency, against different kinds of online policy learning approaches. Additionally, BCV-LR achieved an average of 79% of expert performance across all tasks, reaching expert levels in many tasks such as "Maze", "Bigfish", "Fruitbot", and "Starpilot". These for the first time demonstrate that videos can support extremely sample-efficient visual policy learning, without the need of accessing expert actions or rewards.

### 4.3 Continuous Control

In this section, we provide the comparison on 8 continuous visual tasks from DMControl. On each task, only 100k environmental steps (50k interactions with action repeat set to 2) are allowed. For ILV methods, we employed a well-trained RL agent (1M steps) to collect 100k transitions as expert

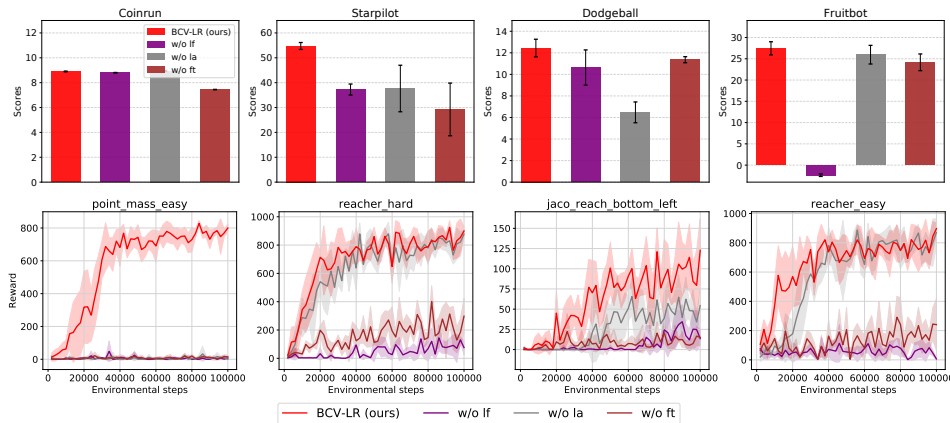

Figure 4: Ablation study on both discrete control and continuous control.

videos. The RL methods are permitted to access the environmental rewards. Refer to Appendix C for detailed hyper-parameter settings.

We first provide the comparison between ILV methods and show their training curves in Figure 3. For the inverse RL method LAIFO, both reward prediction and value estimation require advanced understanding of the environments. It can deal well with continuous control when given enough interactions but can't perform well with limited samples. Compared with the other supervised ILV methods, BCV-LR still exhibits huge advantages, which is similar to results in discrete control. In Table 5, we further provide the numerical evaluation results of sample-efficient visual RL baselines that are provided with ground-truth rewards. Compared with these advanced RL methods, BCV-LR is still better on 6/8 tasks at 100 steps. In addition, BCV-LR is able to learn effective policies across all tasks at only 50k environmental steps (even 20k for some tasks), which RL baselines cannot achieve (refer to Appendix B). In summary, the results in this section further verify (i) the superiority of the proposed BCV-LR and (ii) the potential of videos as solitary supervision.

In addition to DMControl, we also conduct experiments on 4 continuous manipulation tasks from the Metaworld benchmark. Refer to Appendix B for results and details.

## 4.4 Ablation Study

In this section, we ablate each key component of BCV-LR to show the effect on the ILV performance, as shown in Figure 4. In "BCV-LR w/o lf", we give up self-supervised visual pre-training (corresponding to Section 3.1.1), unfreezing the feature encoder $f$ and updating it via substantial training objectives. This leads to consistent performance drops, especially in continuous control, demonstrating that a steady visual representation is crucial for understanding complex dynamics. In "BCV-LR w/o la", we don't pre-train latent actions (corresponding to Section 3.1.2) but learn predictor $p$ and world model $w$ in the online stage, observing clear sample efficiency differences. Particularly for "point_mass_easy", effective learning was unattainable without latent actions, highlighting the significance of pre-trained knowledge. Finally, we fix the pre-trained latent actions, only updating the decoder $d$ to align the latent action space and real action space online (corresponding to Section 3.2.1), denoted as "BCV-LR w/o ft". The drops are not large in Procgen but extremely huge in continuous control with more complex dynamics, which is consistent with the failure of baseline UPESV that freezes the pre-trained knowledge. In conclusion, the results demonstrate that each module is necessary for BCV-LR to achieve sample-efficient ILV performance in both discrete control and continuous control. In Appendix B, we further provide more ablation results about different sub-optimization objectives introduced in Section 3.1.1 and 3.2.1.

## 4.5 Video Data Efficiency

In this chapter, we give BCV-LR different numbers of demonstration videos to explore its data efficiency, and the results are shown in Figure 5. 50k transitions are enough for BCV-LR to achieve performance next to the expert level across both tasks. When given only 20k transitions, BCV-LR can still learn the expert-level policy on "finger_spin" and an effective policy on "reacher_hard". While

BCV-LR has demonstrated a certain degree of data efficiency, it exhibits performance bottlenecks when the amount of video data is further reduced to 5k. On the basis of high sample efficiency, how to further improve the video data efficiency of BCV-LR is another issue worthy of research.

### 4.6 Multi-task Pre-training and Adaptation

In previous experiments, BCV-LR's workflow is completed on a single task: offline pre-training on expert videos of one single task, followed by online finetuning and policy cloning in the reward-free environment corresponding to the task. In this chapter, we attempt to explore the multi-task potential of BCV-LR. Following the multi-task pre-training settings employed in previous works [51, 34], we pre-train one BCV-LR model on mixed videos of three diverse tasks ('Bigfish', 'Maze', and 'Starpilot') and then achieve online finetuning in these tasks and two unseen tasks ('Bossfight' and 'Dodgeball') separately. The results in Table 3 demonstrate that the multi-task BCV-LR enables effective policy imitation on all

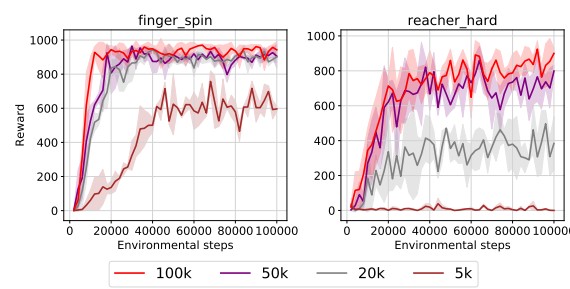

Figure 5: The training curves of BCV-LR when given different numbers of action-free video transitions. 50k video transitions are enough for BCV-LR to learn an effective policy.

tasks, achieving robust multi-task pre-training and cross-domain adaptation. The pre-trained knowledge can be shared across both seen and unseen domains, demonstrating the potential of BCV-LR to leverage large-scale cross-domain video data from the internet.

## 5 Limitation & Insight

Despite the remarkable results across several challenging tasks, BCV-LR faces the same issues as all methods based on behavior cloning: covariate shift [25]. This issue restricts the ability to handle sequential decision-making tasks, such as complex robot locomotion. We provide additional experimental analysis in Appendix B. Some recent works have demonstrated the feasibility of combining inverse RL and behavior cloning to address state-based ILO problems [65]. To this end, integrating BCV-LR with existing inverse rewards or designing inverse rewards based on BCV-LR latent representations are both viable improvement directions, which we leave for future work. Additionally, directions such as attempting to further improve BCV-LR's video data efficiency or exploring its ability to leverage large-scale videos from the internet also represent future research avenues.

Table 3: Multi-task pre-training and adaptation of BCV-LR. BCV-LR-M denotes the variant of BCV-LR with multi-task pre-training. PPO-S and BCV-LR-S denote training under the default single-task setting, i.e., they are the same as those in Section 4.2.

| Task | BCV-LR-M | BCV-LR-S | PPO-S[63] | Expert Videos |
|---|---|---|---|---|
| Bigfish | 32.2 ± 2.0 | 35.9 ± 2.0 | 0.9 ± 0.1 | 36.3 |
| Maze | 9.6 ± 0.1 | 9.9 ± 0.1 | 5.0 ± 0.7 | 10.0 |
| Starpilot | 44.3 ± 1.9 | 54.8 ± 1.4 | 2.6 ± 0.9 | 67.0 |
| Bossfight | 5.5 ± 0.3 | 10.3 ± 0.3 | 0.1 ± 0.1 | 11.6 |
| Dodgeball | 9.5 ± 0.3 | 12.4 ± 0.8 | 1.1 ± 0.2 | 13.5 |

## 6 Conclusion

In this paper, we propose a novel framework, named BCV-LR, to efficiently derive policies from action-free videos without accessing rewards or any other expert supervision. Extensive experiments demonstrate the huge efficiency advantages of BCV-LR against both advanced ILV methods and RL methods. BCV-LR for the first time demonstrates the feasibility of using videos as the only expert supervisory signal to guide extremely sample-efficient policy learning, and we believe this work can serve as a key stepping stone towards sample-efficient ILV in more scenarios (e.g., real-world robot manipulation) where both traditional supervision and environmental interactions are expensive.

## Acknowledgments and Disclosure of Funding

This work is partly supported by the National Natural Science Foundation of China (NSFC) under Grants No. 62173324 and No. 62206281, and in part by the CAS for Grand Challenges under Grant 104GJHZ2022013GC, and in part by the Suzhou Innovation and Entrepreneurship Leading Talents Programme - Innovation Leading Talent in Universities and Research Institutes with Grant No. ZXL2025310.

The authors would like to thank Yaran Chen (XJTLU), Boyu Li (CASIA), and Zhennan Jiang (CASIA) for their help and discussions on this work.

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

# APPENDIX

## A  Supplemented Details of BCV-LR Methodology

### A.1  Pseudo Code

---

**Algorithm 1** The pseudo code of the proposed BCV-LR.

---

**Require:** The action-free video dataset $B^v$, the reward-free environment $E$, the latent feature pre-training update times $U_{lf}$, latent action pre-training steps $U_{la}$, the online environmental interaction steps $I_{ft}$, the online finetuning frequency $F_{ft}$, the finetuning update times $U_{ft}$, the behavior cloning update times $U_{bc}$.

**Initialize:** The latent feature encoder $f$, the latent action predictor $p$, the world model $w$, the latent action decoder $d$, the latent policy $\pi$, and the environmental replay buffer $B^e$.

1:   *# Offline stage #*
2:   *## Pre-training latent features ##*
3:   **for** $index = 1, ..., U_{lf}$ **do**
4:       Sample a batch of video observations from video dataset $B^v$.
5:       Calculate the self-supervised objective $\mathcal{L}_{lf}$ via Equation (1).
6:       Update the latent feature encoder $f$ by backpropagation.
7:   *## Pre-training latent actions ##*
8:   **for** $index = 1, ..., U_{la}$ **do**
9:       Sample a batch of video observations from video dataset $B^v$.
10:      Calculate the dynamics-based objective $\mathcal{L}_{la}$ via Equation (2).
11:      Update the latent action predictor $p$ and the world model $w$ by backpropagation.
12:   *# Online stage #*
13:   **for** $index = 1, ..., I_{ft}/F_{ft}$ **do**
14:      *## Optional preparations ##*
15:      (Optional, letting latent policy $\pi$ fully imitate the pre-trained latent actions before interactions.)
16:      **if** $index == 1$ :
17:         Conduct latent behavior cloning via Equation (4) until convergence.
18:      *## Collect environmental interactions ##*
19:      **for** $index' = 1, ..., F_{ft}$ **do**
20:         **if** $index == 1$ :
21:            Interact with environment $E$ to collect data with random policy.
22:            Save the data into the buffer $B^e$
23:         **else**:
24:            Interact with environment $E$ to collect data with the learned policy (consisting of $f$,$\pi$, and $d$).
25:            Save the data into the buffer $B^e$
26:      *## Finetuning and decoding the latent actions ##*
27:      **for** $index' = 1, ..., U_{ft}$ **do**
28:         Sample a batch of environmental transitions from buffer $B^e$.
29:         Calculate the latent action finetuning objective $\mathcal{L}_{ft}$ via Equation (3).
30:         Update the latent action predictor $p$ and the latent action decoder $d$ by backpropagation.
31:      *## Behavior cloning the latent policy ##*
32:      **for** $index' = 1, ..., U_{bc}$ **do**
33:         Sample a batch of action-free observations from videos $B^v$.
34:         Obtain the predicted latent actions through $f$ and $p$.
35:         Calculate the latent behavior cloning objective $\mathcal{L}_{bc}$ via Equation (4).
36:         Update the latent policy $\pi$.

**Output:** The well-trained visual control policy ($f$,$\pi$,and $d$).

---

## A.2 Prototype-based Temporal Association Task

As shown in Section 3.1.1, for partially observable domains with complex dynamics, such as DMControl benchmark [31] in this paper, we employ a recent advanced prototype-based temporal association task [57, 43, 34], which aligns two temporally neighboring observations together in the latent space. For the sake of fluency, we put the details of this self-supervised task here instead of in the main text.

Concretely, $M$ observation pairs $\{(o_i^v, o_{i+1}^v)\}$ are sampled from videos, augmented by the random shift, and encoded by $f$ and $f'$ to obtain latent features $\{s_i^v\}$ and $\{s_{i+1}^v\}$. Each current latent feature $s_i^v$ is further processed by an MLP $u$, which is set to introduce asymmetry for avoiding collapse to trivial solutions. Then, we take a softmax over the dot product between $u(s_i^v)$ and $M$ trainable prototypes $\{c_j\}_{j=1}^M$:

$$x_i^v = \text{softmax}\left(\frac{u(s_i^v) \cdot c_1}{\tau}, ..., \frac{u(s_i^v) \cdot c_M}{\tau}\right), \tag{5}$$

where $\tau$ denotes a temperature hyper-parameter. To calculate the association target, the Sinkhorn-Knopp [61] algorithm is employed on the whole batch $\{s_{i+1}^v\}$ and prototypes $\{c_j\}$ to obtain batch-clustering target labels $\{y_{i+1}^v\}$ for all training feature pairs. Concretely, the Sinkhorn-Knopp algorithm begins with the square matrix $C$, whose elements are computed by the dot product over each $s_{i+1}^v$ and prototype $c_j$:

$$C_{ij} = s_{i+1}^v \cdot c_j. \tag{6}$$

Then it employs several times of doubly-normalization on the matrix $C$ to obtain the clustering target matrix $T$, constraining every column and row to have the same sum with as little change of original $C$ as possible. One doubly-normalization consists of a row normalization and a column normalization. The row normalization and column normalization are formulated as the following:

$$\text{NormRow}(C) = \frac{1}{M}\text{diag}(\text{SumRow}(C)^{-1}) \cdot C, \tag{7}$$

$$\text{NormColumn}(C) = \frac{1}{M}C \cdot \text{diag}(\text{SumColumn}(C)^{-1}), \tag{8}$$

where $\text{SumRow}(\cdot)$ denotes the row addition, $\text{SumColumn}(\cdot)$ denotes the column addition, and $\text{diag}(\cdot)$ denotes the diagonalization of a matrix. The doubly-normalization, $\text{NormDouble}(\cdot)$, is correspondingly defined as:

$$\text{NormDouble}(C) = \text{NormColumn}(\text{NormRow}(C)). \tag{9}$$

Several times of doubly-normalization are applied to $C$ to obtain the clustering target matrix $T$. The $i$-th row of $T$ is the clustering-based target $y_{i+1}^v$ of the $x_i^v$. The self-supervised objective is formulated as:

$$L_{lf} = -{y_{i+1}^v}^\top \log x_i^v. \tag{10}$$

The clustering-based temporal association task actually makes each observation access its own temporally neighboring observation in the self-supervised latent space, thus enabling the representation module $f$ to obtain the temporal information.

# B Additional Results

## B.1 Numerical Results at Fewer Steps in DMControl

In Section 4.3, we provide the results at 100k environmental steps, where the proposed BCV-LR leads across 6/8 tasks. Here we demonstrate the advantages of BCV-LR when interactions are further constrained (50k and 20k steps). As shown in Table 8 and Table 9, BCV-LR performs much better than both ILV and RL baselines, enabling effective policy learning given only 20k steps.

Table 4: Numerical results at 50k environmental steps. *Italics* indicate using expert videos, and underlines denote using environmental rewards. **Bold text** indicates the highest score excluding video experts.

| Task (50k steps) | *BCV-LR (ours)* | *LAIFO* [50] | *BCO* [26] | *UPESV* [28] | TACO [60] | DrQv2 [3] | Expert Videos |
|---|---|---|---|---|---|---|---|
| point_mass_easy | **743 ± 76** | 1 ± 0 | 3 ± 4 | 3 ± 4 | 529 ± 85 | 107 ± 143 | 885 |
| reacher_hard | **860 ± 64** | 22 ± 33 | 261 ± 142 | 14 ± 6 | 41 ± 44 | 4 ± 3 | 967 |
| jaco_reach_top_right | **73 ± 13** | 14 ± 14 | 0 ± 0 | 3 ± 3 | 10 ± 8 | 22 ± 16 | 198 |
| finger_spin | **912 ± 38** | 166 ± 122 | 855 ± 28 | 0 ± 0 | 416 ± 41 | 165 ± 121 | 981 |
| ball_in_up_catch | **683 ± 106** | 25 ± 43 | 208 ± 227 | 33 ± 47 | 126 ± 115 | 176 ± 120 | 986 |
| jaco_reach_bottom_left | **101 ± 32** | 20 ± 22 | 0 ± 0 | 8 ± 4 | 8 ± 6 | 12 ± 10 | 203 |
| cheetah_run_backward | 268 ± 8 | 1 ± 0 | 207 ± 27 | 3 ± 1 | **272 ± 157** | 220 ± 158 | 389 |
| reacher_easy | **747 ± 101** | 80 ± 17 | 110 ± 123 | 3 ± 4 | 110 ± 15 | 202 ± 152 | 975 |

Table 5: Numerical results at 20k environmental steps. *Italics* indicate using expert videos, and underlines denote using environmental rewards. **Bold text** indicates the highest score excluding video experts.

| Task (20k steps) | *BCV-LR (ours)* | *LAIFO* [50] | *BCO* [26] | *UPESV* [28] | TACO [60] | DrQv2 [3] | Expert Videos |
|---|---|---|---|---|---|---|---|
| point_mass_easy | **318 ± 219** | 1 ± 1 | 0 ± 0 | 0 ± 0 | 1 ± 0 | 1 ± 1 | 885 |
| reacher_hard | **714 ± 46** | 22 ± 36 | 69 ± 85 | 20 ± 14 | 26 ± 35 | 8 ± 8 | 967 |
| jaco_reach_top_right | **70 ± 22** | 5 ± 5 | 6 ± 8 | 4 ± 3 | 2 ± 1 | 3 ± 2 | 198 |
| finger_spin | **944 ± 25** | 84 ± 84 | 799 ± 82 | 0 ± 0 | 51 ± 68 | 43 ± 59 | 981 |
| ball_in_up_catch | **459 ± 139** | 96 ± 121 | 196 ± 215 | 99 ± 81 | 99 ± 81 | 33 ± 47 | 986 |
| jaco_reach_bottom_left | **45 ± 28** | 1 ± 1 | 0 ± 1 | 8 ± 6 | 3 ± 3 | 2 ± 2 | 203 |
| cheetah_run_backward | **226 ± 11** | 1 ± 0 | 188 ± 14 | 5 ± 1 | 150 ± 94 | 67 ± 48 | 389 |
| reacher_easy | **569 ± 206** | 77 ± 32 | 131 ± 92 | 0 ± 0 | 136 ± 59 | 38 ± 31 | 975 |

## B.2 Experiments in Metaworld Manipulation

In this section, we further conduct some extra experiments in the Metaworld manipulation benchmark [32], which may demonstrate a wider application of BCV-LR. For each Metaworld task, only 50k environmental steps are allowed. The remaining settings are similar to that of DMControl experiments, except for the self-supervised latent feature objective $\mathcal{L}_{lf}$ employed in BCV-LR. Concretely, we employ only contrastive learning loss (described in Section 3.1.1) in this benchmark because it seems to perform better than other objectives used in this paper. Results (success rate) are shown in Table 6. In this interaction-limited situation, BCV-LR can still derive effective manipulation skills from expert videos without accessing expert actions and rewards, which demonstrates its wider range of applications and potential for generalizing to real-world manipulation tasks.

Table 6: The results (success rate) on 4 manipulation tasks from Metaworld. *Italics* indicate using expert videos, and underlines denote using environmental rewards. **Bold text** indicates the highest score excluding video experts.

| Metaworld-50k | *BCV-LR (ours)* | *BCO* [26] | DrQv2 [3] | Expert Videos |
|---|---|---|---|---|
| Faucet-open | **0.82 ± 0.20** | 0.13 ± 0.19 | 0.00 ± 0.00 | 1.00 |
| Reach | **0.63 ± 0.25** | 0.03 ± 0.05 | 0.13 ± 0.12 | 1.00 |
| Drawer-open | **0.92 ± 0.12** | 0.13 ± 0.09 | 0.00 ± 0.00 | 1.00 |
| Faucet-close | **0.98 ± 0.04** | 0.00 ± 0.00 | 0.50 ± 0.28 | 1.00 |
| Mean Success Rate | **0.84** | 0.07 | 0.16 | 1.00 |

## B.3 Additional Comparison with LAPO's BC Variant

In Section 4, we use LAPO [21] as a reinforcement learning baseline, which follows its original paper and official implementation. In this section, we replace LAPO's online RL loss with behavior cloning (BC) loss, obtaining its reward-free BC variant, and then compare it with BCV-LR. The results in Table 7 show that LAPO-BC works well in some tasks, but our BCV-LR still performs better.

Table 7: The comparison between BCV-LR and LAPO's BC variant. *Italics* indicate using expert videos, and underlines denote using environmental rewards. **Bold text** indicates the highest success rate excluding video experts.

| Task | *BCV-LR (ours)* | *LAPO-BC* [21] | PPO [63] | Expert Videos |
|---|---|---|---|---|
| Fruitbot | **27.5 ± 1.5** | 6.2 ± 1.9 | -1.9 ± 1.0 | 29.9 |
| Heist | **9.3 ± 0.1** | 9.2 ± 0.3 | 3.7 ± 0.2 | 9.7 |
| Bossfight | **10.3 ± 0.3** | 0.0 ± 0.0 | 0.1 ± 0.1 | 11.6 |
| Chaser | **3.1 ± 0.5** | 0.6 ± 0.0 | 0.4 ± 0.2 | 10.0 |

## B.4 Ablation&Hyper-parameter Sensitivity: Self-supervised Reconstruction Loss

In this section, we further provide the ablation study of the extra reconstruction loss employed in Section 3.1.1. In video games, useful visual information is abundant and scattered. Unlike contrastive learning, which understands the image as a whole, the reconstruction task forces the feature encoder to focus on these key pixels. The results demonstrate that this extra objective can indeed improve the performance, as long as it is not set too large, which is shown in Figure 6.

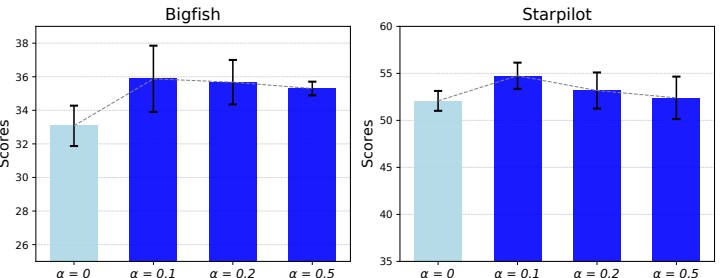

Figure 6: Ablation and hyper-parameter sensitivity analysis of the extra reconstruction loss in self-supervised latent feature pre-training.

## B.5 Ablation&Hyper-parameter Sensitivity: Finetuning Latent Actions with World Model

In this section, we demonstrate whether finetuning the latent actions online with the world model yields better performance. The results in Figure 7 indicate that constraining the latent action finetuning with the pre-trained world model is helpful. It is consistent with our intuition, because we aim to extract the expert actions for policy cloning, while the online collection is not expert-level in most of the time. The pre-trained world model can provide the expert dynamics-based knowledge to alleviate the effect of the distribution difference.

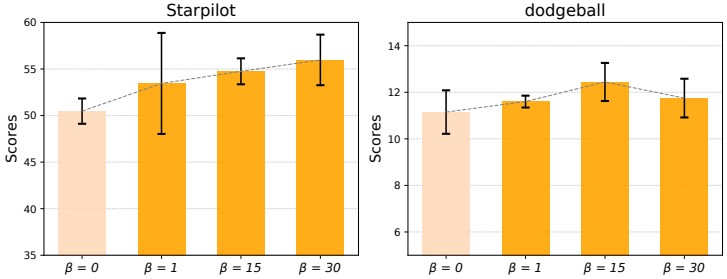

Figure 7: Ablation and hyper-parameter sensitivity analysis of the dynamics-constrained loss in online latent action finetuning.

## B.6 Ablation: Freezing Latent Features

In this section, we provide ablation experiments to demonstrate whether it is necessary to further fine-tune the pre-trained self-supervised latent features with the other objectives in BCV-LR. We finetune the self-supervised encoder with $\mathcal{L}_{la}$ and $\mathcal{L}_{ft}$, respectively. The results in Table 8 demonstrate that whether finetuning self-supervised visual features doesn't yield an apparent effect on policy performance. This phenomenon has also been observed in self-supervised RL [59, 57], leading some works to fine-tune self-supervised features while others opt to freeze them.

Table 8: Comparison of policy learning performance between freezing and fine-tuning BCV-LR self-supervised latent features.

| Task | Finetuned via $\mathcal{L}_{la}$ | Finetuned via $\mathcal{L}_{ft}$ | Frozen | Expert videos |
|---|---|---|---|---|
| reacher_hard | 876 ± 15 | **906 ± 65** | 900 ± 31 | 967 |
| finger_spin | 937 ± 26 | 920 ± 57 | **942 ± 48** | 981 |

## B.7 Schedule of Environment Interactions

In the online stage of BCV-LR, a) we allow the agent to interact with the environment for a fixed number of times using its policy and collect transitions to enrich the experience buffer. Immediately after that, b) we perform finetuning of the latent action and training of the action decoder on the experience buffer. Then, c) we train the latent policy to imitate the finetuned latent action predicted from expert videos. After this, the BCV-LR policy is improved, and we return to part a) to collect better training data, which forms a cyclic online policy learning. Under the default experimental settings, we set the number of interactions for each cycle at a relatively large value from start to finish (for example, we fixed the number of collected transitions in part a) as 1000 in DMControl). In this section, we try a much smaller interaction number (set to 2) and correspondingly reduce the number of update times for latent actions (set to 1) and latent policies (set to 2) in each cycle. We denote this variant as 'BCV-LR(1000->2)'. The results in Table 9 show that our approach can still achieve effective policy learning, demonstrating its robustness to the schedule of environment interactions.

Table 9: Comparison of BCV-LR's performance under different schedules of environment interactions.

| Task | BCV-LR (1000->2) | BCV-LR | DrQv2[3] | Expert videos |
|---|---|---|---|---|
| reacher_hard | 875 ± 65 | **900 ± 31** | 92 ± 98 | 967 |
| finger_spin | **956 ± 20** | 942 ± 48 | 374 ± 264 | 981 |

## B.8 Offline Latent Action Decoding

In this section, we let BCV-LR perform latent action finetuning and policy imitation with a few offline action-labeled expert transitions, in a fashion akin to LAPA [19]. Concretely, we maintain the original pre-training stage and use 10k offline action-labeled expert transitions to achieve offline latent action finetuning and policy cloning with the original losses. The results in Table 10 (RL denotes DrQv2 [3] for DMControl tasks and PPO [63] for Procgen tasks) demonstrate that BCV-LR can also achieve offline imitation learning well if expert actions are provided.

Table 10: BCV-LR can also accomplish latent action decoding and fine-tuning using offline action-labeled expert data.

| Task | BCV-LR-offline | BCV-LR | RL | Expert videos |
|---|---|---|---|---|
| reacher_hard | **938 ± 44** | 900 ± 31 | 92 ± 98 | 967 |
| finger_spin | **978 ± 7** | 942 ± 48 | 374 ± 264 | 981 |
| Fruitbot | **27.7 ± 0.4** | 27.5 ± 1.5 | -1.9 ± 1.0 | 29.9 |

## B.9 Limitation Analysis

Despite the remarkable results across different kinds of control tasks (including both discrete control and continuous control), BCV-LR faces covariate shift [25], the same issues as all methods based on behavior cloning. This issue restricts the ability to handle sequential decision-making tasks, such as complex robot locomotion. In this section, we provide the comparison between the IRL method LAIFO [50] and our BCV-LR on two continuous tasks: the balance control "reacher_hard" and the locomotion task "walker_walk". As shown in Figure 8, when the environmental interactions are limited (only 100k steps), BCV-LR exhibits significant advantages on both tasks.

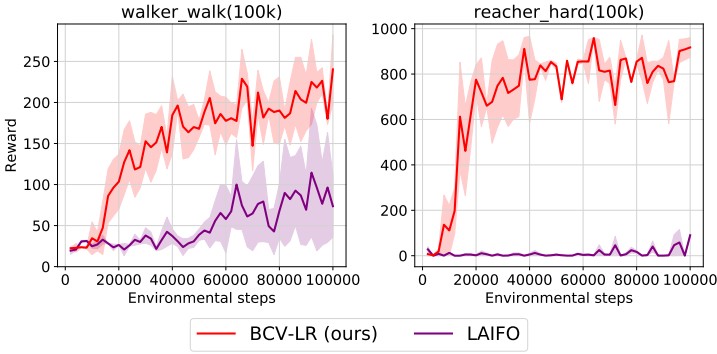

Figure 8: The training curves of BCV-LR and LAIFO when 100k steps are allowed. BCV-LR exhibits advantages on both locomotion and balance control.

However, if enough environmental steps (500k) are permitted, behavior cloning-based BCV-LR's performance growth stagnates after efficient learning in the sequential decision-making task "walker_walk", while RL-based LAIFO can continue to improve.

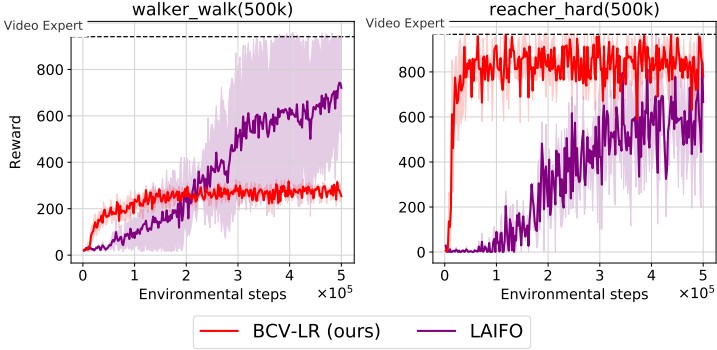

Figure 9: The training curves of BCV-LR and LAIFO when 500k steps are allowed. BCV-LR is limited in sequential decision-making task, which is due to the covariate shift of the behavior cloning.

Some recent works have demonstrated the feasibility of combining inverse RL and behavior cloning to address state-based ILO problems [65]. To this end, integrating BCV-LR with existing inverse rewards or designing inverse rewards based on BCV-LR latent representations are both viable improvement directions, which we leave for future work.

# C  Experimental Details

## C.1  Introductions of Environments

In this paper, we employ diverse and challenging visual control tasks to provide a comprehensive evaluation of the proposed BCV-LR, as shown in Figure 10. The top two rows are discrete Procgen tasks [30], the third row contains continuous DMControl tasks [31], while the bottom row contains Metaworld manipulation tasks [32].

The first row from left to right is Coinrun, Starpilot, Caveflyer, Dodgeball, Leaper, Maze, Bigfish, and Heist.

The second row from left to right is Fruitbot, Chaser, Miner, Jumper, Climber, Plunder, Ninja, and Bossfight.

The third row from left to right is jaco_reach_bottom_left, point_mass_easy, finger_spin, reacher_easy, cheetah_run_backward, ball_in_cup_catch, jaco_reach_top_right, and reacher_hard.

The bottom row from left to right is Faucet-open, Reach, Drawer-open, and Faucet-close.

See their paper for the detailed task description.

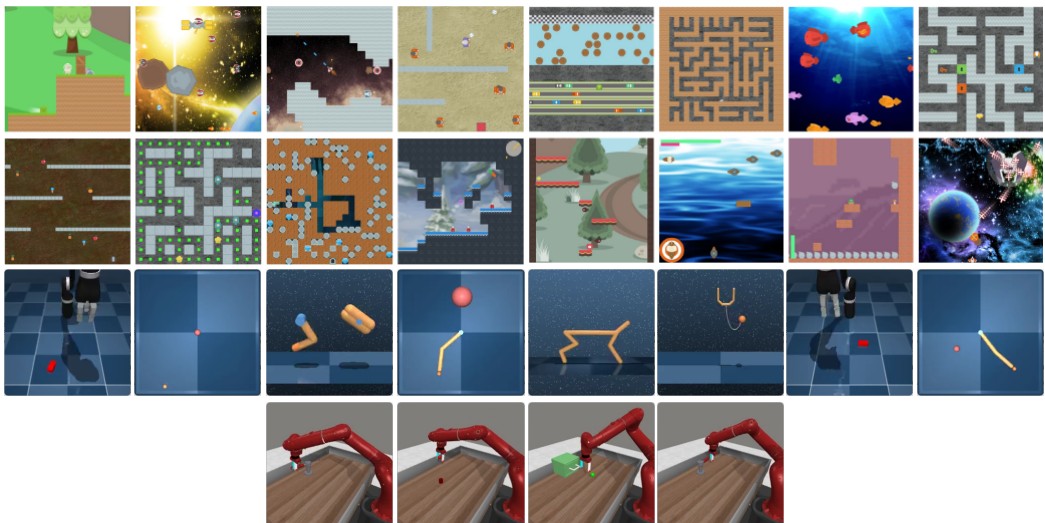

Figure 10: Screenshot of diverse and challenging visual control tasks employed in this paper.

## C.2 Hyper-parameter Settings

Table 11: Default hyper-parameter settings in discrete Procgen. The "lr" denotes "learning rate", while the "ft" denotes "finetuning". There are additionally 60k latent behavior cloning update times before the first round of the online stage to ensure that the latent policy fully imitates the pre-trained latent actions. For Miner, Climber, and Chaser, the "online latent action predictor ft lr" is 1e-3 and the $\beta$ is set to 1. For Plunder, the $\alpha$ is 1. The experiments of BCV-LR are conducted using V100 or A800 GPUs, and the complete workflow for each task can be finished within five hours on a single GPU.

| Hyper-parameter | Setting |
|---|---|
| Frame shape | $64 \times 64 \times 3$ |
| Frame stack | 1 |
| Action repeat | 1 |
| Action type | Discrete |
| Action dimension | 15 |
| Video steps | 8M |
| Video expert training steps | 50M |
| Latent feature pt times | 20000 |
| Feature encoder lr | $3e-5$ |
| Random shift padding upper bound | 1 |
| Reconstruction coefficient $\alpha$ | $1e-1$ |
| EMA update frequency | 2 |
| EMA Momentum | 0.05 |
| VQ codebook number | 2 |
| VQ dicrete latent number | 4 |
| VQ latent embedding dimension | 16 |
| VQ latent embedding number | 64 |
| Latent action dimension | 128 |
| Latent action pt times | 10000 |
| Latent action predictor pt lr | $3e-4$ |
| World model pt lr | $3e-4$ |
| Online steps | 100000 |
| Online env numbers | 64 |
| Online update frequency | 64 |
| Latent action ft update times | 60 |
| Latent action decoder lr | $1e-3$ |
| Latent action predictor ft lr | $2e-5$ |
| World model loss coefficient $\beta$ | 15 |
| World model ft | Frozen |
| Behavior cloning update times | 500 |
| Latent policy lr | $2e-4$ |

Table 12: Default hyper-parameter settings in DMControl. The "lr" denotes "learning rate", while the "ft" denotes "finetuning". There are additionally 60k latent behavior cloning update times before the first round of the online stage to ensure that the latent policy fully imitates the pre-trained latent actions. For 'point_mass_easy', the random shift padding is up to 1. For the domain jaco and reacher, the update time of latent action is set to the fixed value of $100$ while the $\beta$ and the world model finetuning learning rate are both $1e - 3$. For Metaworld tasks, they share hyper-parameters with DMControl, except for the action dimension set to 4, the online interaction frames set to 50k, and the latent feature self-supervised task changed to contrastive learning. The experiments of BCV-LR are conducted using V100 or A800 GPUs, and the complete workflow for each task can be finished within five hours on a single GPU.

| Hyper-parameter | Setting |
|---|---|
| Frame shape | $64 \times 64 \times 3$ |
| Frame stack | 3 |
| Action repeat | 2 |
| Action type | Continuous |
| Action dimension | Refer to [31] |
| Video frames | 200k |
| Video expert training frames | 1M |
| Temperature hyper-parameter $\tau$ | 0.1 |
| Latent feature pt times | 50000 |
| Feature encoder lr | $1e - 4$ |
| Random shift padding upper bound | 4 |
| Numbers of doubly-normalization | 3 |
| EMA update frequency | 2 |
| EMA Momentum | 0.05 |
| VQ codebook number | 2 |
| VQ dicrete latent number | 4 |
| VQ latent embedding dimension | 16 |
| VQ latent embedding number | 64 |
| Latent action dimension | 128 |
| Latent action pt times | 50000 |
| Latent action predictor pt lr | $3e - 4$ |
| World model pt lr | $3e - 4$ |
| Online interaction frames | 100000 |
| Online env numbers | 1 |
| Online update frequency | 1000 |
| Latent action ft update epoch | 2 |
| Latent action decoder lr | $1e - 3$ |
| Latent action predictor ft lr | $1e - 3$ |
| World model loss coefficient $\beta$ | $1e - 5$ |
| World model ft lr | $1e - 5$ |
| Behavior cloning update times | 1000 |
| Latent policy lr | $1e - 3$ |

