# OpenReview forum: "Videos are Sample-Efficient Supervisions: Behavior Cloning from Videos via Latent Representations"
_NeurIPS.cc/2025/Conference — NeurIPS 2025 poster_

### Official Review · Reviewer_TLPv · 2025-06-24

**Clarity:** 4
**Significance:** 2
**Originality:** 3
**Rating:** 5
**Confidence:** 3

**Summary:**

This paper introduces a method for sample-efficient imitation learning from videos. This works by first training an encoder on videos by using a contrastive loss between the frame and a shifted version of the frame, as well as a temporal loss for the next frame feature similarity. This paper also uses a world model to learn a latent action representation, which is then mapped to a real world action using a few expert trajectories. The paper evaluates on video-game tasks and compares to many prior works.

**Questions:**

How general is the learned encoder? Is the shift image + contrastive loss learn useful features for non-video games, say real world robotics tasks? Or will this only work for relatively simplistic videos/environments?

How well does the mapping from latent action to action work in more complex environments, e.g., robot controls rather than just video game controls?

**Ethical Concerns:**

["NO or VERY MINOR ethics concerns only"]

**Final Justification:**

The rebuttal addressed my concerns. I think it is a good paper with meaningful contributions.

**Limitations:**

Yes

**Quality:**

3

**Strengths And Weaknesses:**

The paper is well written, easy to understand, and presents an approach to a important problem. The results are good, showing more sample efficient learning than prior works. The idea is interesting, and learning from videos without requiring action annotations is a hard problem.


There are a few weaknesses, mostly in that the training of the encoder seems limiting, and I have doubts it will generalize to more complex and realistic environments. Same for the mapping from the latent actions to real actions. The main novelty of the approach is in taking these parts and making them work for the task, ie. using the contrastive pre-training + world model + latent action + action mapping. So if these methods don't generalize to real world tasks, it will greatly limit the impact of the paper.

---

> ### Author Rebuttal · Authors · 2025-07-31
>
> We greatly appreciate the reviewer's careful reading, constructive reviews, and strong recognition of our work! We have carefully reviewed your comments, addressed your questions, and further improved our paper based on your feedback!
> ___
> >**Q1: How general is the learned encoder? Is the shift image + contrastive loss learn useful features for non-video games, say real world robotics tasks? Or will this only work for relatively simplistic videos/environments?**
>
> We apply this ‘shift image + contrastive loss’ with reconstruction (Eq.1, line 185, main paper) for Procgen video games because it has been proven effective in these kinds of tasks [1,2]. As shown in Sec.3.1.1 (line 174, main paper), BCV-LR is easily compatible with any action-free self-supervised tasks, and it can adapt to different types of domains by choosing appropriate self-supervised objectives. This motivates us to choose another prototype-based temporal association loss (Eq.10, line 549, Appendix) for DMControl environments where the temporal understanding is crucial and this loss has been proven better than ‘shift image + contrastive loss’ in DMControl [3].
>
> To this end, BCV-LR can involve more advanced self-supervised objectives (e.g., ViT-based masked reconstruction [4,5]) for more challenging tasks (e.g., real-world manipulation) if necessary. In addition, BCV-LR can also be combined with an off-the-shelf, well-trained encoder, which makes its potential not limited to only video game environments.
>
>
> ___
> >**Q2: How well does the mapping from latent action to action work in more complex environments, e.g., robot controls rather than just video game controls?**
>
> In addition to the video games, we also demonstrate the advantages of BCV-LR on DMControl benchmark which consists of several continuous robotic control tasks. We summarize the average results below. More details are provided in Sec.4.3 (main paper) and Sec.C.1 (Appendix).
>
> **Table I**
> | DMControl-8-tasks-100k  | BCV-LR  | LAIFO   | BCO  |UPESV |TACO  | DrQv2| / |Expert video|
> | - | - | - | - | - | - |  - | - | - |
> | Mean Score | **604**   | 158  | 336  | 18     | 310  | 232 | | 698|
> | Video-norm Mean Score  | **0.78**   | 0.20  | 0.31  | 0.03     | 0.45  | 0.34 | | 1.00|
>
>
> In addition, we further conduct additional experiments in the Metaworld manipulation benchmark. Only 50k environmental steps are allowed for each Metaworld task, with remaining settings similar to that of DMControl. Results (success rate) are shown in Table II. In this interaction-limited situation, BCV-LR can still derive effective manipulation skills from expert videos without accessing expert actions and rewards, which demonstrates its wider range of applications and potential for generalizing to real-world manipulation tasks.
>
> **Table II**
> |Metaworld-50k|BCV-LR|BCO|DrQv2|/|Expert video|
> |-|-|-|-|-|-|
> |Faucet-open|**0.82 ± 0.20**|0.13 ± 0.19|0.00 ± 0.00||1|
> |Reach|**0.63 ± 0.25**|0.03 ± 0.05|0.13 ± 0.12||1|
> |Drawer-open|**0.92 ± 0.12**|0.13 ± 0.09|0.00 ± 0.00||1|
> |Faucet-close|**0.98 ± 0.04**|0.00 ± 0.00|0.50 ± 0.28 ||1|
> |Mean SR|**0.84**|0.07|0.16||1|
>
>
>
> ___
>
> **Thanks for your careful reading and for recommending our paper!**
>
>
> [1]Become a Proficient Player with Limited Data through Watching Pure Videos. ICLR 2023
>
> [2]Curl: Contrastive unsupervised representations for reinforcement learning. ICML 2020
>
> [3]Reinforcement learning with prototypical representations. ICML 2021
>
> [4]Masked world models for visual control. CoRL 2023
>
> [5]Mask-based Latent Reconstruction for Reinforcement Learning. Neurips 2022

---

### Official Review · Reviewer_HHys · 2025-07-01

**Clarity:** 3
**Significance:** 3
**Originality:** 3
**Rating:** 5
**Confidence:** 4

**Summary:**

The paper proposes a method for Imitation Learning from Videos, that is, the problem of learning a policy from videos of expert demonstrations with no action information. To do this, the authors propose "Behavior Cloning from Videos via Latent Representations" (BCV-LR).

The method: 1. learns latent features from video through self-supervision over video frames; 2. then learns a separate world model on the latent features, together with a latent action model; 3. online finetunes the latent action model and a learns a latent action decoder; 4. simultaneously to (3.), trains a latent policy that uses the latent action decoder to act.

The authors conduct experiments across discrete control and continuous control benchmarks, comparing their method to other ILV and RL based baselines.

**Questions:**

* As above: how exactly are phases 3 and 4 optimally alternated? Does this matter?
* How does the method compare to performing the steps in a fashion akin to LAPA? (No online learning, only imitation learning alignment on ground truth expert action?

**Ethical Concerns:**

["NO or VERY MINOR ethics concerns only"]

**Final Justification:**

The in-depth rebuttal further strengthens the paper's claim, I will thus maintain my acceptance score.

**Limitations:**

yes

**Quality:**

3

**Strengths And Weaknesses:**

Strengths:

* The paper focuses on an important and timely problem, as imitation learning becomes widespread state of the art for many real world robotics applications in manipulation and beyond. Many are attempting to learn from large scale video, as it promises to be an extremely cheap and scalable data source of scarce robotics data.
* The paper operates with state of the art methods in representation learning (self-supervision, latent actions, etc) and provides additional evidence towards a "representation-learning first" approach to solving traditional RL tasks and benchmarks. Specifically, the method's performance rivals or beats that of pure RL methods that have access to environmental rewards.
* The comparisons with state of the art methods are extensive (within the chosen eval suite) and prove that the method beats alternatives.

Weaknesses:
* The overall method is complex, consisting of many stages that all have to work for the method to achieve its performance. The ablation study does however demonstrate that each component is required.
* Stage 3. (as per the summary above), in which the latent action model is finetuned and the action decoder is trained, requires real actions to be observed via environment interaction. It is unclear what is the impact of the schedule of environment interactions, how exactly the phases are optimally alternated so as to sample iteratively from the policy learned online via the concurrent step 4. Does this matter?
* The method as requested does require a source of ground truth actions from the videos, it is thus not strictly suitable for solving pressing hard problems for real world robotics, e.g. learning from real human videos and then transferring the policies to robots. This is also related to the benchmarks being limited to classic "toy RL" simulation environments.
* How does the method compare to performing the steps in a fashion akin to LAPA? (No online learning, only imitation learning alignment on ground truth expert action?

---

> ### Author Rebuttal · Authors · 2025-07-31
>
> We are deeply grateful to the reviewer for your careful reading, constructive reviews, and strong recommendation of our paper! We have further enhanced our paper in light of your feedback by supplementing more experiments and discussions!
>
> ___
> >**Weakness1：The overall method is complex, consisting of many stages that all have to work for the method to achieve its performance. The ablation study does however demonstrate that each component is required.**
>
> Thank you for your recognition of our experimental section. We agree with your opinion that the methodology section of BCV-LR is relatively complex. This complexity arises because achieving sample-efficient online policy learning from videos without access to expert actions or rewards is an extremely challenging task. To address this, BCV-LR incorporates multiple training stages, which allow it to obtain as much useful information as possible from both expert videos and environmental samples, which ultimately contributes to its promising results.
>
> ___
> >**Weakness2&Q1：Stage 3. (as per the summary above), in which the latent action model is finetuned and the action decoder is trained, requires real actions to be observed via environment interaction. It is unclear what is the impact of the schedule of environment interactions, how exactly the phases are optimally alternated so as to sample iteratively from the policy learned online via the concurrent step 4. Does this matter?**
>
> The BCV-LR online stage contains three parts. First, we (a) allow the agent to interact with the environment for a fixed number of times using its policy and collect transitions to enrich the experience buffer. Immediately after that, (b) we perform finetuning of the latent action and training of the action decoder on the experience buffer. Then, (c) we train the latent policy to imitate the finetuned latent action predicted from expert videos. After this, the BCV-LR policy is improved and we return to part a) to collect better training data, which forms a cyclic online policy learning. This alternation is intuitive, that is, first collect new data of higher quality, then use the better data to obtain the better latent actions from expert videos, and finally let the policy learn the better latent actions to achieve performance improvement. You can refer to Sec.A.1 (Appendix) for concrete pseudo code of BCV-LR and we will further clarify the descriptions in the methodology section to make them more comprehensible to readers.
>
> **For the impact of the schedule of environment interactions.** In previous experiments, we did not adjust the number of interactions for each cycle, but kept it fixed at a relatively large value from start to finish (for example, we fixed the number of collected transitions in part (a) as 1000 in DMControl) and ultimately achieved satisfactory results. To answer your question, we conduct additional experiments, where we try a much smaller interaction number (1000->2) and correspondingly reduce the number of update times for latent actions (100->1) and latent policies (1000->2) in each interaction, making BCV-LR perform in a fashion akin to off-policy RL. We denote this variant as 'BCV-LR(1000->2)'. The results in Table I show that BCV-LR can still achieve effective learning, which demonstrates that BCV-LR is robust to the schedule of environment interactions.
>
> **Table I**
> | task  | BCV-LR(1000->2) |  BCV-LR |   DrQv2| / |Expert video|
> | - | - | - | - | - | - |
> | reacher_hard |  875 ± 65 |  **900 ± 31**  | 92 ± 98  | | 967 |
> | finger_spin  | **956 ± 20**  | 942 ± 48  | 374 ± 264 || 981|
>
>
>
> ___
> >**Weakness3: The method as requested does require a source of ground truth actions from the videos, it is thus not strictly suitable for solving pressing hard problems for real world robotics, e.g. learning from real human videos and then transferring the policies to robots. This is also related to the benchmarks being limited to classic "toy RL" simulation environments.**
>
> We understand your concerns! Since it is highly challenging to balance video imitation performance and efficiency without access to expert actions or expert rewards, our experiments have primarily been conducted in relatively standard visual RL environments to answer the open question "is sample efficient ILV available" and have not been extended to real-world tasks or internet-scale pre-training, which are of greater significance. Taking your comment into account, we further conduct some extra experiments in Metaworld manipulation benchmark, which may demonstrate a wider application of BCV-LR for robots.
>
> For each Metaworld task, only 50k environmental steps are allowed, with remaining settings similar to that of DMControl. Results (success rate) are shown in Table II. In this interaction-limited situation, BCV-LR can still derive effective manipulation skills from expert videos without accessing expert actions and rewards, which demonstrates its wider range of applications and potential for generalizing to real-world manipulation tasks.
>
> **Table II**
> |Metaworld-50k|BCV-LR|BCO|DrQv2|/|Expert video|
> |-|-|-|-|-|-|
> |Faucet-open|**0.82 ± 0.20**|0.13 ± 0.19|0.00 ± 0.00||1|
> |Reach|**0.63 ± 0.25**|0.03 ± 0.05|0.13 ± 0.12||1|
> |Drawer-open|**0.92 ± 0.12**|0.13 ± 0.09|0.00 ± 0.00||1|
> |Faucet-close|**0.98 ± 0.04**|0.00 ± 0.00|0.50 ± 0.28 ||1|
> |Mean SR|**0.84**|0.07|0.16||1|
>
> For cross-domain video data (e.g., human videos), one challenge BCV-LR may face stems from the difficulty of transferring visual knowledge. Considering that BCV-LR is designed to be easily compatible with any action-free self-supervised tasks, this issue could potentially be alleviated by involving more advanced self-supervised objectives or using off-the-shelf pre-trained encoders. We have added the above discussion in the future work section to inspire more thinking.
>
>
> ___
> >**Weakness4&Q2: How does the method compare to performing the steps in a fashion akin to LAPA? (No online learning, only imitation learning alignment on ground truth expert action?**
>
> To answer your questions, we further conduct additional experiments, where we let BCV-LR perform latent action finetuning and policy imitation with a few action-labeled expert transitions. Concretely, we maintain the original pre-training stage and use 10k offline action-labeled expert transitions to achieve offline latent action finetuning and policy cloning with the original losses. The results in Table III (RL denotes DrQv2 for DMControl and PPO for procgen) demonstrate that BCV-LR can also achieve offline imitation learning well if expert actions are provided.
>
> **Table III**
> || task  | BCV-LR-offline |  BCV-LR |  RL | / |Expert video|
> |-| - | - | - | - | - | - |
> |DMControl| reacher_hard |  **938 ± 44** |  900 ± 31  | 92 ± 98  | | 967 |
> || finger_spin  | **978 ± 7**  | 942 ± 48  | 374 ± 264 || 981|
> |Procgen| fruitbot  | **27.7 ± 0.4**  | 27.5 ± 1.5  | -1.9 ± 1.0 || 29.9|
>
>
> Of course, we would like to say that BCV-LR is designed for the ILV (imitation learning from videos) problem, a much harder variant of the classical ILO (imitation learning from observation only) problem, where the employment of unsupervised online policy training without accessing expert actions adheres to the norms of this field.
>
> ___
> **Thanks for your careful reading and for recommending our paper!**

---

> > ### Comment · Reviewer_HHys · 2025-08-05
> >
> > Thank you very much for the in-depth response, I think that any conceivable weakness has been thoroughly addressed. I will thus keep recommending acceptance.

---

> > > ### Author Response · Authors · 2025-08-06
> > > **Thank you for your reply and we really appreciate your strong recommendation!**
> > >
> > > We are delighted that our response has satisfied you and addressed your concerns!
> > >
> > > Once again, we deeply appreciate the time you invested, your professional feedback, and your strong recommendation for our paper!

---

### Official Review · Reviewer_vqWk · 2025-07-02

**Clarity:** 3
**Significance:** 2
**Originality:** 2
**Rating:** 4
**Confidence:** 3

**Summary:**

In this paper, it presents a new framework, Behavior Cloning from Videos via Latent Representations (**BCV-LR**), that performs imitation learning directly from expert videos without access to rewards or action labels. The method works in two stages: (1) an offline pretraining stage that learns a latent representation of video frames via first self-supervised representation learning and then extracts latent actions through a dynamics-based objective, and (2) an online finetuning phase where latent actions are aligned with real actions using minimal interactions in a reward-free environment. Experimental results demonstrate that BCV-LR achieves superior sample efficiency and outperforms state-of-the-art imitation learning and reinforcement learning baselines across 24 visual control tasks, including both discrete (Procgen) and continuous (DMControl) settings.

**Questions:**

1. For continuous control tasks, they are relatively easy with low-dimensional action spaces. I'm curious how would the proposed methods work for environments with high-dimensional action spaces. (For example, take humanoid & dog from Deepmind Control Suite)
2. The author mention that during self-supervised representation learning phase, it also learns some temporal dynamics by aligning representation of $o_t$ with $o_{t+1}$, but this seems not appearing in Equation 1. So what's the overall objective for this stage. Is it Equation 1 (contrastive loss with image reconstruction ) plus temporal loss?
3. The reconstruction loss of feature learning seems suboptimal especially for environments with distracting backgrounds. The authors mention that they get better performance with this loss added. But is it because the visual observations of the environments it tests on are too simple?
4. A very relevant work [1] is missing.

[1] Ye et al. Become a Proficient Player with Limited Data through Watching Pure Videos, ICLR 2023

**Ethical Concerns:**

["NO or VERY MINOR ethics concerns only"]

**Final Justification:**

This paper is well-motivated with great empirical performance demonstrated in its experiments. During the rebuttal process, the author has addressed my concern of low-dimensional action space and they have also added experiments of Metaworld to demonstrate the method's broader applicability. Therefore, I recommend accept for this paper.

**Limitations:**

Yes

**Quality:**

3

**Strengths And Weaknesses:**

**Strengths**:
1. The paper presentation is good and easy to follow
2. Good motivation to learn from action-free videos
3. Good empirical performance with great sample efficiency: the proposed BCV-LR framework demonstrates impressive sample efficiency in both discrete and continuous control tasks. It performs competitively with or surpasses methods that have access to ground-truth rewards or expert actions.

**Weaknesses**:
The scope of the experiments are relatively limited. All experiments are conducted in simulation/"toy" environments (Procgen and DMControl). While these are standard benchmarks, it is unclear how well BCV-LR generalizes to internet-scale human videos and real-world robotics applications.

---

> ### Author Rebuttal · Authors · 2025-07-31
>
> We greatly appreciate the reviewer’s careful reading, detailed feedback, and recommendation of our paper! We have carefully reviewed your comments and further refined our paper based on your constructive reviews!
>
> ___
> >**Weakness: The scope of the experiments are relatively limited. All experiments are conducted in simulation/"toy" environments (Procgen and DMControl). While these are standard benchmarks, it is unclear how well BCV-LR generalizes to internet-scale human videos and real-world robotics applications.**
>
> We understand your concern! Since it is highly challenging to balance video imitation performance and efficiency without access to expert actions or expert rewards, our experiments have primarily been conducted in relatively standard visual RL environments to answer the open question "is sample efficient ILV available" and have not been extended to real-world tasks or internet-scale pre-training, which are of greater significance. Based on your feedback, we further conduct some extra experiments in Metaworld manipulation benchmark, which may demonstrate a wider application of BCV-LR.
>
> For each Metaworld task, only 50k environmental steps are allowed, with remaining settings similar to that of DMControl. Results (success rate) are shown in Table I. In this interaction-limited situation, BCV-LR can still derive effective manipulation skills from expert videos without accessing expert actions and rewards, which demonstrates its wider range of applications and potential for generalizing to real-world manipulation tasks.
>
>
>
> **Table I**
> |Metaworld-50k|BCV-LR|BCO|DrQv2|/|Expert video|
> |-|-|-|-|-|-|
> |Faucet-open|**0.82 ± 0.20**|0.13 ± 0.19|0.00 ± 0.00||1|
> |Reach|**0.63 ± 0.25**|0.03 ± 0.05|0.13 ± 0.12||1|
> |Drawer-open|**0.92 ± 0.12**|0.13 ± 0.09|0.00 ± 0.00||1|
> |Faucet-close|**0.98 ± 0.04**|0.00 ± 0.00|0.50 ± 0.28 ||1|
> |Mean SR|**0.84**|0.07|0.16||1|
>
> For internet-scale cross-domain video data, one challenge BCV-LR may face stems from the difficulty of transferring visual knowledge. Considering that BCV-LR is designed to be easily compatible with any action-free self-supervised tasks, this issue could potentially be alleviated by involving more advanced self-supervised objectives or using off-the-shelf pre-trained encoders. We have added the above discussion in the future work section to inspire more thinking.
>
> ___
> >**Q1:For continuous control tasks, they are relatively easy with low-dimensional action spaces. I'm curious how would the proposed methods work for environments with high-dimensional action spaces. (For example, take humanoid & dog from Deepmind Control Suite)**
>
> Following your suggestion, we attempt experiments on the 'dog' and 'humanoid' domains. We find that these two domains are highly challenging even for advanced RL methods that use expert rewards. Specifically, we first try training agents to collect expert data via RL, but observe that both TACO and DrQ fail within 5M steps (obtaining scores lower than 5). Furthermore, we run DrQv2 on the 'humanoid_walk' task for 20M steps across 4 seeds, with only 1 run successfully learning a policy. We use this 20M-step policy to collect data, and then train BCV-LR and all baselines under the 100k-step setting—all of which failed. (Table II) This indicates that complex dynamics and action spaces remain significant challenges for current policy learning algorithms, even when expert rewards are provided. Meanwhile, BCV-LR, which is designed to balance sample efficiency under more challenging settings (without access to rewards or expert actions), is not yet able to handle such tasks effectively.
>
> **Table II**
> | Task-100k | BCV-LR  | LAIFO   | BCO  |UPESV |TACO  | DrQv2| / |Expert video|
> | - | - | - | - | - | - |  - | - | - |
> | humanoid_walk |1.3 ± 0.1  | 1.1 ± 0.0 | 1.1 ± 0.1 | 1.0 ± 0.1 | 2.0 ± 0.2 | 1.9 ± 0.2| |529|
>
> ___
> >**Q2:The author mention that during self-supervised representation learning phase, it also learns some temporal dynamics by aligning representation of $o_t$ with $o_{t+1}$, but this seems not appearing in Equation 1. So what's the overall objective for this stage. Is it Equation 1 (contrastive loss with image reconstruction ) plus temporal loss?**
>
> As shown in Sec.3.1.1 (line 174, main paper), BCV-LR is designed to be easily compatible with any action-free self-supervised tasks (it completely decouples visual learning from subsequent training), and it can adapt to different types of tasks by choosing appropriate self-supervised objectives. To this end, we apply different latent feature training losses for Procgen and DMControl. Concretely, we employ the 'contrastive loss with image reconstruction' (Eq.1, line 185, main paper) for Procgen video games, because it has been proven effective in these kinds of tasks [1,2]. For DMControl environments where the temporal understanding is crucial, we choose another prototype-based temporal association loss (Eq.10, line 549, Appendix) because involving temporal information into self-supervised objectives have been proven necessary in DMControl [3]. In summary, BCV-LR can involve more advanced self-supervised objectives for more challenging tasks if necessary, which makes its potential not limited to specific environments.
>
> We have further improved the presentation in Section 3.1.1 based on the above details to enhance clarity for readers.
>
> ___
> >**Q3:The reconstruction loss of feature learning seems suboptimal especially for environments with distracting backgrounds. The authors mention that they get better performance with this loss added. But is it because the visual observations of the environments it tests on are too simple?**
>
> We agree with your opinion on reconstruction loss, and some previous works [6] have demonstrated the limitation of reconstruction when faced with visual distraction. As we explain in our answer to Q2, BCV-LR employs the reconstruction loss in Procgen because of its effectiveness in video games [1], but it is not limited to this reconstruction loss. BCV-LR is designed to be easily compatible with any action-free self-supervised tasks (it completely decouples visual learning from subsequent training), and it can adapt to different types of tasks by choosing appropriate self-supervised objectives. To this end, BCV-LR can involve more advanced self-supervised objectives (e.g., ViT-based masked reconstruction [4,5] or distraction-robust prototypical representation [6]) for more challenging tasks if necessary. In addition, BCV-LR can also be combined with an off-the-shelf, well-trained encoder, which makes its potential not limited to both tasks and experimental settings.
>
> ___
> >**Q4:A very relevant work [1] is missing.**
>
> We appreciate your careful reading, and now have supplemented the discussion of FICC [1] in the related work (line 121): "FICC[47] pre-trains world models on action-free agent experience, accelerating MBRL on Atari." Thank you!
>
>
>
> ___
>
> **We hope these answers can meet your expectations, and we would be most grateful if you maintain your recommendation or raise your score. Thanks!**
>
> [1]Become a Proficient Player with Limited Data through Watching Pure Videos. ICLR 2023
>
> [2]Curl: Contrastive unsupervised representations for reinforcement learning. ICML 2020
>
> [3]Reinforcement learning with prototypical representations. ICML 2021
>
> [4]Masked world models for visual control. CoRL 2023
>
> [5]Mask-based Latent Reconstruction for Reinforcement Learning. Neurips 2022
>
> [6]Dreamerpro: Reconstruction-free model-based reinforcement learning with prototypical representations. ICML 2022

---

> > ### Comment · Reviewer_vqWk · 2025-08-03
> >
> > Thanks the authors for the clarifications. I think they have mainly addressed my concern. I would like to keep my score and recommend acceptance for this paper.

---

> > > ### Author Response · Authors · 2025-08-04
> > > **Thank you for your reply and your recommendation for our paper!**
> > >
> > > We are very pleased that our response has effectively addressed your concerns! Once again, we appreciate the time you’ve dedicated, your constructive reviews, and your recommendation for the acceptance!

---

### Official Review · Reviewer_oQMP · 2025-07-02

**Clarity:** 3
**Significance:** 2
**Originality:** 3
**Rating:** 4
**Confidence:** 4

**Summary:**

The work proposes BCV-LR to conduct behavior cloning with expert videos and several online interactions. BCV-LR learns the dynamics model and inverse dynamics model on latent actions during the offline phase. In the online phase, it achieves behavior cloning of expert videos by aligning latent actions to true actions through interaction. Without action labels, experiments conducted on Procgen and DMControl demonstrate BCV-LR  with good sample efficiency.

**Questions:**

1.  Does the phrase "without the need to access any other supervision" include the supervision of actions during the online phase? The supervision on the actions directly injects the environment's inverse dynamics knowledge and potentially adjusts the distribution of latent actions through alignment with ground-truth actions. If the phrase refers to merely using the latent reward information implicit in expert videos without online expert rewards for video behavior cloning, JEPT[1] may be an uncited but related work. Similarly, without using an explicit reward signal, JEPT can achieve generalization of one-shot visual imitation in some tasks with a mixture dataset.
2.  The design of BCV-LR is very similar to FICC[2] (mainly in the offline part), while FICC is not properly referenced. FICC should be a comparable baseline in the discrete setting.
3.  I am curious whether the effectiveness of direct behavior cloning in the online phase is related to the single-task setting or the quality of expert data. The offline phase of BCV-LR is quite similar to LAPO and FICC, while both LAPO and FICC avoid direct behavior cloning in the online phase and choose to use online rewards for policy adjustment in the multitask setting. It would be better if a direct online BC variant of LAPO were involved in the comparison, which means the online reward will also be excluded in this setting, and may further illustrate this aspect.
4.  The paper lacks experiments in a multi-task setting or a discussion on the relationship between the expert video scale and the performance in the single-task setting. Can BCV-LR generalize to new tasks through multi-task data? Or, for new tasks lacking sufficient data, how much offline data is required for BCV-LR to fulfill behavior cloning? Otherwise, the prerequisite of obtaining sufficient data often implies having a well-performing in-domain policy on the task, reducing the significance of behavior cloning from expert videos.
5.  Adding some ablation experiments would be more helpful in explaining what factors make BCV-LR more effective than previous works. For example, a) the impact on performance of whether the representation learning part is directly frozen after pre-training or updated with $L_{la}$ or $L_{ft}$; b) the impact on performance of not updating the latent action predictor in the online phase.

[1] Learning Video-Conditioned Policy on Unlabelled Data with Joint Embedding Predictive Transformer. Hao Luo, Zongqing Lu. ICLR 2025.

[2] Become a proficient player with limited data through watching pure videos. Weirui Ye, Yunsheng Zhang, Pieter Abbeel, and Yang Gao. ICLR 2023.

**Ethical Concerns:**

["NO or VERY MINOR ethics concerns only"]

**Final Justification:**

The author's rebuttal addresses most of my concerns. Importantly, the weakened version of the claim mitigates my concerns about overclaiming.

**Limitations:**

yes

**Paper Formatting Concerns:**

No formatting issues.

**Quality:**

2

**Strengths And Weaknesses:**

Strengths:
The presentation of the paper is quite clear. The experimental performance of BCV-LR is excellent, especially considering that only a small number of online steps are used.

Weaknesses:
I am not sure whether the offline stage is individually conducted on each task or jointly conducted on a multi-task dataset. The practical significance of video behavior cloning in a single-task setting is limited. The LAPO in the baseline is designed for the pre-training on multi-task data, which may affect comparisons. Additionally, incorporating extra ablation studies and baselines could further solidify this work. Please refer to the Questions part.

---

> ### Author Rebuttal · Authors · 2025-07-31
>
> We appreciate your careful reading, professional comments, and affirmation of our paper! In response to your feedback, we have supplemented the missing related works and further conducted several extra experiments, which greatly improves the paper quality. We hope these revisions will meet your expectations, and we would be most grateful if you could consider raising your score!
> ___
> >**W1:I am not sure whether the offline stage is individually conducted on each task or jointly conducted on a multi-task dataset. The practical significance of video behavior cloning in a single-task setting is limited.**
>
> In the submitted version, all experiments are conducted under single-task settings. This is mainly due to two reasons: (a) whether it is possible to balance video imitation performance and sampling efficiency in single-task settings remained an open problem before our paper; and (b) recent advanced pure ILV (no reward) works mainly achieve evaluation under single-task settings. We agree with you that achieving video cloning under multi-task settings is more meaningful and we supplement multi-task experiments as you suggested, as shown in answer to Q4.
> ___
> >**W2:The LAPO in the baseline is designed for the pre-training on multi-task data, which may affect comparisons.**
>
> LAPO indeed inspires many works on multitask pre-training. However, according to the final published version of LAPO and its open-source code, it uses a 'full' distribution for each task in Procgen to ensure intra-task diversity (e.g., the 'maze' task includes mazes of different sizes, colors, and layouts) while doesn't perform cross-task 'multi-task learning' (i.e., it achieves pre-training on 'maze' videos and then performs online learning in the 'maze' task). The Procgen experiments of BCV-LR have strictly followed LAPO's setup in terms of environment configuration, and we use the same video dataset as that of LAPO, thus ensuring a fair comparison. Please feel free to let us know if you have any further questions!
> ___
> **Due to space constraints, we can't list your questions here. Please refer to your reviews. We apologize for the inconvenience.**
> >**To Q1: We supplement the reference and discussion of JPET.**
>
> Thank you for your careful reading! BCV-LR utilizes information from environmental actions. Our intention is to emphasize that it doesn't rely on 'expert' actions. We have replaced "without the need to access any other supervision" with "without the need to access any other expert supervision" to eliminate ambiguity.
>
> In addition, JPET[1] is very relevant to our work and we missed it. We have added the citation and discussion of JPET in the related work (line 120): "JPET[46] utilizes a mixed dataset containing videos to achieve one-shot visual imitation."
> ___
> >**To Q2: We add the reference and discussion of FICC and explain why it has not been included in the comparison for the time being.**
>
> Thanks for your careful reading, and we now have discussed FICC [2] in the related work (line 121): "FICC[47] pre-trains world models on action-free agent experience, accelerating MBRL on Atari." That said, due to the following reasons, we have not yet included FICC in the experiments on ProcGen:
>
> 1. FICC has not been evaluated on the Procgen and none of our baselines evaluates FICC on Procgen. Moreover, whether its MCTS-based backbone EfficientZero (EZ)[3] is applicable to procedurally generated environments remains an open question. This lack of information—such as hyperparameter settings and EZ framework in Procgen—leaves us unable to ensure the validity of our reproductions.
> 2. We have already compared BCV-LR against 6 popular baselines on Procgen, including the state-of-the-art RL and ILV algorithms. FICC is not able to perform policy learning without relying on rewards, in the way that BCV-LR can. Its training paradigm is more similar to that of LAPO. Given that LAPO itself has achieved state-of-the-art performance on ProcGen, we believe direct comparisons with LAPO may be more persuasive.
> 3. FICC provides code for its pre-training part while doesn't offer code for fine-tuning the pre-trained model and combining it with its MCTS-based backbone online. This makes it difficult for us to reproduce this work within the short timeframe of the rebuttal period.
> 4. The backbone of FICC, EZ[3], is built on MCTS and thus requires considerable training time. As noted in EZ, it takes 28 GPU hours for 100k steps in discrete control, whereas BCV-LR online stage requires only around 1 GPU hour on both discrete and continuous domains. This high computational cost has led many subsequent works [4,5] to refrain from direct comparisons with EZ, even in Atari experiments. In addition, this consumption also means it's hard to obtain its results within the limited timeframe available.
> ___
> >**To Q3: We explain that the single-task setting and training data don't cause unfairness. We conduct extra experiments on LAPO's BC variant.**
>
> As we explain in our answer to W2, LAPO also focuses on single-task settings. FICC includes multi-task pre-training experiments, but its main results are also single-task.​
> What’s more, the Procgen video we use is the same as LAPO’s original data, as stated in Sec.4.2 (line286). This means all methods have expert data of the same quality.​
>
> As you asked, we conducted extra experiments: we replaced LAPO’s online RL loss with BC loss, then compared it with BCV-LR. The results show that LAPO-BC works well in some tasks, but our BCV-LR still performs better.​
> We also show that BCV-LR can be used for multi-task pre-training in a fashion akin to FICC. Please refer to answer to Q4.
> |Task|BCV-LR|LAPO-BC|PPO|/|Expert video|
> |-|-|-|-|-|-|
> |fruitbot|**27.5 ± 1.5**|6.2 ± 1.9|-1.9 ± 1.0||29.9|
> |heist|**9.3 ± 0.1**|9.2 ± 0.3|3.7 ± 0.2||9.7|
> |bossfight|**10.3 ± 0.3**|0.0 ± 0.0|0.1 ± 0.1||11.6|
> |chaser|**3.1 ± 0.5**|0.6 ± 0.0|0.4 ± 0.2||10.0|
> ___
> >**To Q4: We conduct extra experiments on multi-task ability of BCV-LR and provide video data efficiency analysis.**
>
> According to your suggestions, we conducted extra multitask pre-training experiments for BCV-LR, following the settings of that in FICC. We pre-train one BCV-LR model on mixed videos of 'bigfish', 'maze', and 'starpilot', then finetune it in these tasks separately. Different from FICC, we further employ two unseen tasks. The results show that BCV-LR enables effective policy imitation on all tasks. It achieves robust multitask pre-training, where the pre-trained knowledge can be shared across both seen and unseen domains.
> ||Task|BCV-LR(multi)|PPO(single)|/|BCV-LR(single)|
> |-|-|-|-|-|-|
> |seen|bigfish|**32.2 ± 1.0**|0.9 ± 0.1||35.9|
> ||maze|**9.6 ± 0.1**|5.0 ± 0.7||9.9|
> ||starpilot|**44.3 ± 1.9**|2.6 ± 0.9||54.8|
> |unseen|bossfight|**5.5 ± 0.3**|0.1 ± 0.1||10.3|
> ||dodgeball|**9.5 ± 0.3**|1.1 ± 0.2||12.4|
>
> Then, we provide the video data efficiency experiments. We test BCV-LR with 5k, 20k, 50k, and 100k (default setting) video transitions. Results demonstrate that BCV-LR enables effective policy learning with only 20k transitions. 50k transitions can support near-expert performance. Refer to Appendix C.5 for more details.
> |Video data of BCV-LR|5k|20k|50k|100k|/|Expert video|
> |-|-|-|-|-|-|-|
> |reacher_hard|0 ± 0|384 ± 153|799 ± 34|**900 ± 31**||967|
> |finger_spin|596 ± 17|901 ± 33|905 ± 70|**942 ± 48**||981|
> ___
> >**To Q5: We conduct extra ablation experiments.**
>
> As per your suggestions, we first conducted additional experiments, where we finetuned the self-supervised encoder with $L_{la}$ and $L_{ft}$ respectively. The results in Table IV demonstrate that whether finetuning self-supervised visual representation doesn't yield apprent effct on policy performance (curves are also similar). This phenomenon has also been observed in self-supervised RL[7], leading some works to fine-tune self-supervised features while others opt to freeze them.
> |Task|Finetuning via $L_{la}$|Finetuning via $L_{ft}$|No visual finetuning|
> |-|-|-|-|
> |reacher_hard|876 ± 15|**906 ± 65**|900 ± 31|
> |finger_spin|937 ± 26|920 ± 57|**942 ± 48**|
>
> Then, we provide the experiments to show the impact on performance of not updating the latent action predictor with $L_{ft}$ in the online phase (denoted as 'BCV-LR w/o ft'). Partial results are below, demonstrating that utilizing the environmental actions to finetune pre-trained latent actions is a crucial step in BCV-LR, especially in DMControl tasks. Please refer to Sec.4.4 (main paper) for more results.
> ||Task|BCV-LR|BCV-LR w/o ft|
> |-|-|-|-|
> |DMControl|point_mass_easy|**800 ± 25**|22 ± 2|
> ||jaco_reach_bottom_left|**123 ± 39**|15 ± 10|
> |Procgen|starpilot|**54.8 ± 1.4**|29.2 ± 10.6|
> ||fruitbot|**27.5 ± 1.5**|24.2 ± 2.0|
> ___
> >**Finally, we further conduct additional experiments in Metaworld.**
>
> Only 50k steps are allowed for each task, with other settings similar to that of DMControl. Results (success rate) demonstrate BCV-LR's wider range of applications and potential for generalizing to real-world manipulation tasks.
> |Metaworld-50k|BCV-LR|BCO|DrQv2|/|Expert video|
> |-|-|-|-|-|-|
> |Faucet-open|**0.82 ± 0.20**|0.13 ± 0.19|0.00 ± 0.00||1|
> |Reach|**0.63 ± 0.25**|0.03 ± 0.05|0.13 ± 0.12||1|
> |Drawer-open|**0.92 ± 0.12**|0.13 ± 0.09|0.00 ± 0.00||1|
> |Faucet-close|**0.98 ± 0.04**|0.00 ± 0.00|0.50 ± 0.28||1|
> |Mean SR|**0.84**|0.07|0.16||1|
> ___
> **We hope these revisions can meet your expectations, and we would be most grateful if you consider raising your score. Thanks!**
>
> [1]Learning Video-Conditioned Policy on Unlabelled Data with Joint Embedding Predictive Transformer.ICLR25
>
> [2]Become a Proficient Player with Limited Data through Watching Pure Videos.ICLR23
>
> [3]Mastering Atari Games with Limited Data.Neurips21
>
> [4]Value-Consistent Representation Learning for Data-Efficient Reinforcement Learning.AAAI23
>
> [5]Mask-based Latent Reconstruction for Reinforcement Learning.Neurips22
>
> [6]Learning to act without actions.ICLR24
>
> [7]Decoupling representation learning from reinforcement learning.ICML21

---

> > ### Comment · Reviewer_oQMP · 2025-08-04
> >
> > Thanks for the detailed response, which has addressed most of my concerns.
> >
> > The modified claim fits the setting better.
> >
> > I will raise the score to ‘borderline accept’

---

> > > ### Author Response · Authors · 2025-08-04
> > > **Thank you for your reply. We sincerely appreciate that you have chosen to raise the score！**
> > >
> > > We are very delighted that our response and revisions have addressed most of your concerns! We are also deeply grateful that you have chosen to raise the score to a positive one!
> > >
> > > Once again, thank you for the efforts you’ve dedicated, your professional feedback, and your recommendation for acceptance!

---

### Note · Authors · 2025-08-12

We sincerely thank the four professional reviewers, as well as the ACs and SACs, for their efforts invested in reviewing and processing our paper.

**Contributions of our paper:** Our paper proposes a novel ILV (Imitation Learning from Videos) framework, BCV-LR, to answer an important open question: "Can videos support extremely sample-efficient visual policy learning, without relying on environmental rewards or expert actions?" We conduct extensive evaluations of our method on 28 popular tasks across diverse domains (16 ProcGen tasks, 8 DMControl tasks, and 4 MetaWorld tasks), demonstrating its great sample efficiency advantages over several state-of-the-art online baselines, including both ILV and RL approaches.

**Discussion with reviewers:** During the review phase, the four reviewers unanimously recognized the remarkable effectiveness of the proposed BCV-LR and the clear presentation of the paper, and awarded favorable initial scores. In the rebuttal phase, we provided detailed responses to each of the weaknesses and questions raised by the four reviewers. We are delighted that our answers effectively addressed their main concerns, leading to their consistent recommendation for the acceptance.

Finally, we would like to once again sincerely thank the ACs and SACs for the time they invested in processing our manuscript. We also express our sincere gratitude once more to the four reviewers for their professional feedback and their recommendation of our paper.

---

### Decision · Program_Chairs · 2025-09-17

**Decision:**

Accept (poster)

**Comment:**

The paper proposes BCV-LR, a novel framework for imitation learning from videos (without expert actions but with online interaction) that extracts latent action representations through self-supervised pretraining and aligns them to real action spaces via online interaction. The method demonstrates highly sample-efficient learning, in some cases reaching expert-level performance.

The main concerns are the limited evaluation on relatively simple, short-horizon environments and the complexity of the multi-stage pipeline. The paper also omits discussion of and comparison to a highly relevant prior work [1]. Reviewers found the contribution promising but not fully validated at scale. The rebuttal clarified design choices and provided MetaWorld experiments, but did not address the experimental limitations.

Overall, I lean toward accept: the approach is interesting and results are strong, but more exhaustive ablations and evaluation on realistic tasks and videos would substantially strengthen the paper.


[1] Escontrela et al. "Video Prediction Models as Rewards for Reinforcement Learning" NeurIPS 2023